# Characterization of antimicrobial-resistant Gram-negative bacteria that cause neonatal sepsis in seven low- and middle-income countries

Kirsty Sands[1,2,25]✉, Maria J. Carvalho[1,3,25]✉, Edward Portal[1], Kathryn Thomson[1], Calie Dyer[1,4], Chinenye Akpulu[1,5,6], Robert Andrews[1], Ana Ferreira[1], David Gillespie[4], Thomas Hender[1], Kerenza Hood[4], Jordan Mathias[1], Rebecca Milton[1,4], Maria Nieto[1], Khadijeh Taiyari[4], Grace J. Chan[7,8,9], Delayehu Bekele[9,10], Semaria Solomon[11], Sulagna Basu[12], Pinaki Chattopadhyay[13], Suchandra Mukherjee[13], Kenneth Iregbu[5], Fatima Modibbo[5,6], Stella Uwaezuoke[14], Rabaab Zahra[15], Haider Shirazi[16], Adil Muhammad[15], Jean-Baptiste Mazarati[17], Aniceth Rucogoza[17], Lucie Gaju[17], Shaheen Mehtar[18,19], Andre N. H. Bulabula[19,20], Andrew Whitelaw[21,22], BARNARDS Group[23],* and Timothy R. Walsh[1,24]

Antimicrobial resistance in neonatal sepsis is rising, yet mechanisms of resistance that often spread between species via mobile genetic elements, ultimately limiting treatments in low- and middle-income countries (LMICs), are poorly characterized. The Burden of Antibiotic Resistance in Neonates from Developing Societies (BARNARDS) network was initiated to characterize the cause and burden of antimicrobial resistance in neonatal sepsis for seven LMICs in Africa and South Asia. A total of 36,285 neonates were enrolled in the BARNARDS study between November 2015 and December 2017, of whom 2,483 were diagnosed with culture-confirmed sepsis. *Klebsiella pneumoniae* ($n = 258$) was the main cause of neonatal sepsis, with *Serratia marcescens* ($n = 151$), *Klebsiella michiganensis* ($n = 117$), *Escherichia coli* ($n = 75$) and *Enterobacter cloacae* complex ($n = 57$) also detected. We present whole-genome sequencing, antimicrobial susceptibility and clinical data for 916 out of 1,038 neonatal sepsis isolates (97 isolates were not recovered from initial isolation at local sites). Enterobacterales (*K. pneumoniae, E. coli* and *E. cloacae*) harboured multiple cephalosporin and carbapenem resistance genes. All isolated pathogens were resistant to multiple antibiotic classes, including those used to treat neonatal sepsis. Intraspecies diversity of *K. pneumoniae* and *E. coli* indicated that multiple antibiotic-resistant lineages cause neonatal sepsis. Our results will underpin research towards better treatments for neonatal sepsis in LMICs.

Although there has been a substantial decrease in infant mortality over the past 20 years[1], the burden remains substantial, with the mortality rate of children under 5 years of age at 38 per 1,000 live births in 2019[2], and 98% of recorded neonatal deaths now occurring in LMICs[3]. The World Health Organization (WHO) has declared neonatal sepsis a global concern[4] and the burden of neonatal infectious diseases a major challenge. Effective management of sepsis is not always possible when resources are limited[3], and the steady increase of antimicrobial resistance (AMR) worldwide further compromises sepsis management[5,6].

Despite the burden of neonatal sepsis, accurate information on the causes and consequences of neonatal sepsis in LMICs is scarce[4,6]. Most studies in LMICs are from a single site, are of limited sample size, or lack accurate methods for sepsis diagnosis, pathogen identification and antibiotic susceptibility measurements[7–10]. In 2015 and 2016, two multicentre neonatal sepsis studies in LMICs were published[11,12]. However, neither study combined antimicrobial susceptibility testing and whole-genome sequencing (WGS), making it difficult to determine the extent of genomic diversity (which would usually be done by comparing the lineages across geographical

[1]Division of Infection and Immunity, Cardiff University, Cardiff, UK. [2]Department of Zoology, University of Oxford, Oxford, UK. [3]Institute of Biomedicine, Department of Medical Sciences, University of Aveiro, Aveiro, Portugal. [4]Centre for Trials Research, Cardiff University, Cardiff, UK. [5]National Hospital Abuja, Abuja, Nigeria. [6]54gene, Lagos, Nigeria. [7]Division of Medical Critical Care, Boston Children's Hospital, Boston, MA, USA. [8]Department of Pediatrics, Harvard Medical School, Boston, MA, USA. [9]Department of Epidemiology, Harvard T.H. Chan School of Public Health, Boston, MA, USA. [10]Department of Obstetrics and Gynecology, St Paul's Hospital Millennium Medical College, Addis Ababa, Ethiopia. [11]Department of Microbiology, Immunology and Parasitology, St Paul's Hospital Millennium Medical College, Addis Ababa, Ethiopia. [12]Division of Bacteriology, ICMR–National Institute of Cholera and Enteric Diseases, Kolkata, India. [13]Department of Neonatology, Institute of Postgraduate Medical Education & Research, Kolkata, India. [14]Federal Medical Centre – Jabi, Abuja, Nigeria. [15]Department of Microbiology, Quaid-i-Azam University, Islamabad, Pakistan. [16]Pakistan Institute of Medical Sciences, Islamabad, Pakistan. [17]The National Reference Laboratory, Rwanda Biomedical Centre, Kigali, Rwanda. [18]Unit of IPC, Stellenbosch University, Cape Town, South Africa. [19]Infection Control Africa Network, Cape Town, South Africa. [20]Department of Global Health, Stellenbosch University, Cape Town, South Africa. [21]Division of Medical Microbiology, Stellenbosch University, Cape Town, South Africa. [22]National Health Laboratory Service, Tygerberg Hospital, Cape Town, South Africa. [23]www.barnards-group.com. [24]Ineos Oxford Institute for Antimicrobial Research, Department of Zoology, Oxford, UK. [25]These authors contributed equally: Kirsty Sands, Maria J. Carvalho. *A list of authors and their affiliations appears at the end of the paper. ✉e-mail: kirsty.sands@zoo.ox.ac.uk; mjcarvalho@ua.pt

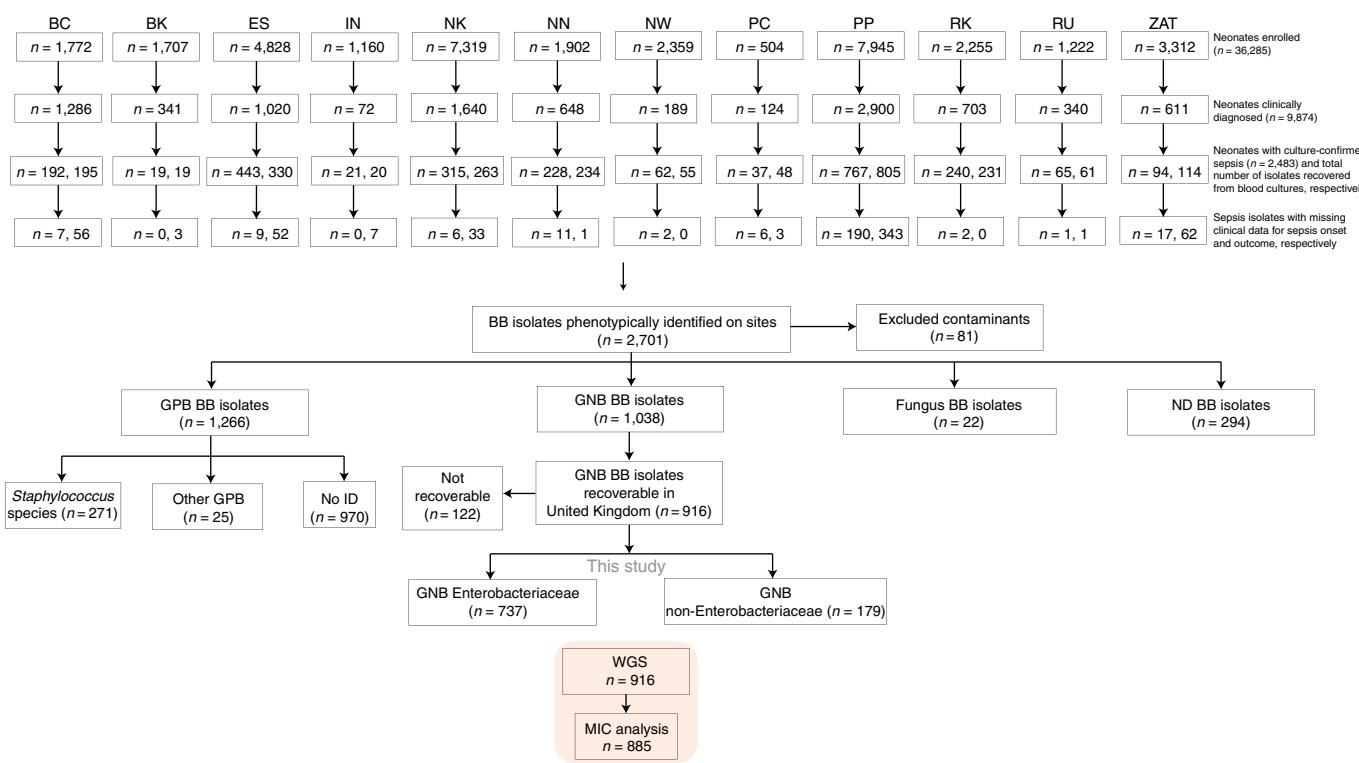

**Fig. 1 | Flow chart detailing the enrolment of neonates and incidence of sepsis (both clinically diagnosed and confirmed by culture) per site.** The numbers of neonates with missing clinical data for the onset and outcome of sepsis are shown per site. The numbers of isolates collected from neonatal blood cultures are shown per site, with a breakdown by preliminary characterization (as determined by Gram stain) in the flow chart below. The final numbers of isolates included for analysis are highlighted in orange. BB, baby blood; ID, identification (of species); ND, not determined.

areas) and resistance. The studies that have taken this approach were of single sites and often used WGS to investigate specific outbreaks[13–16].

In LMICs, the epidemiology of early-onset sepsis (EOS) and late-onset sepsis (LOS) is not well defined[3], unlike in high-income countries, where group B *Streptococcus* is usually considered the main cause of EOS[17]. A systematic review of the causes of blood culture-positive neonatal sepsis in Sub-Saharan Africa by Okomo et al.[18] found that *Klebsiella* species, *Escherichia coli, Enterobacter* species and *Pseudomonas* species accounted for 38% of cases. Other single-site reports showed concordant findings[9,11,12,19]. However, these studies did not specifically determine whether certain species, or sequence type (ST) groups, are more likely to harbour resistance or virulence determinants, how this compares between different geographical areas and whether there is any relation to sepsis onset or outcome.

Burden of Antibiotic Resistance in Neonates from Developing Societies (BARNARDS; www.barnards-group.com) is a network of 12 clinical study sites in four African (Ethiopia, Nigeria, Rwanda and South Africa) and three South Asian countries (Bangladesh, India and Pakistan). The aim of the BARNARDS study is to assess the burden of AMR in neonates in these LMIC. Here, we report on the isolation and characterization of Gram-negative bacteria (GNB) causing neonatal sepsis in seven LMICs, including their AMR profiles. We report associations between phenotypic and genotypic data and sepsis onset and mortality following biological sepsis (MFBS). We also analyse whole-genome sequences from isolates that cause neonatal sepsis.

## Results

**Enrolment in BARNARDS and isolation of pathogenic bacteria.** The numbers of neonates recruited, clinically diagnosed with

sepsis and with a confirmation of sepsis by positive blood culture are outlined in Fig. 1. Of 36,285 infants ≤60 d old (termed herein as neonates) enrolled in the BARNARDS study from November 2015 to December 2017, 2,483 had culture-confirmed sepsis. All 12 clinical sites used the same criteria for clinical diagnosis of sepsis (Supplementary Fig. 1). We found that cases were mainly EOS for both sites in Pakistan, one site in Bangladesh (BC; see Methods for definitions of all two-letter site abbreviations) and the single site in Ethiopia. In Nigeria, India and South Africa, there were mainly LOS cases. In Rwanda, neonatal sepsis cases were equally split between EOS and LOS (Fig. 2).

Automated blood culture systems were used to detect microbial growth, with 2,620 microbial isolates recovered. These 2,620 isolates comprised 1,266 Gram-positive bacteria (GPB) isolates, 1,038 GNB isolates, 22 fungal isolates and 294 unassigned isolates (Fig. 1). The methods for collection and identification were standardized across all sites, with equipment and reagents purchased from uniform suppliers. The primary aim of the BARNARDS study was to characterize the extent of β-lactam resistance in GNB causing clinically diagnosed sepsis in infants <60 d old (see Supplementary Figs. 2 and 3). However, at month 17 (during a BARNARDS network event), we anecdotally noted high rates of isolation of *Staphylococcus* species and therefore collected all of these isolates for analysis (to be reported elsewhere). Of 1,038 isolates, 916 GNB were analysed using WGS. For 122 isolates, identification beyond a Gram stain was not possible because the isolate was lost and/or purification for DNA extraction was unsuccessful (Fig. 1).

In total, 58 different species of GNB were identified across all sites by WGS (Supplementary Table 1), including *K. pneumoniae* (n = 258), *Serratia marcescens* (n = 151), *Klebsiella michiganensis* (n = 117), *Enterobacter* species (n = 80), *E. coli* (n = 75), *Burkholderia* species (n = 61), *Acinetobacter* species (n = 49), *Pseudomonas*

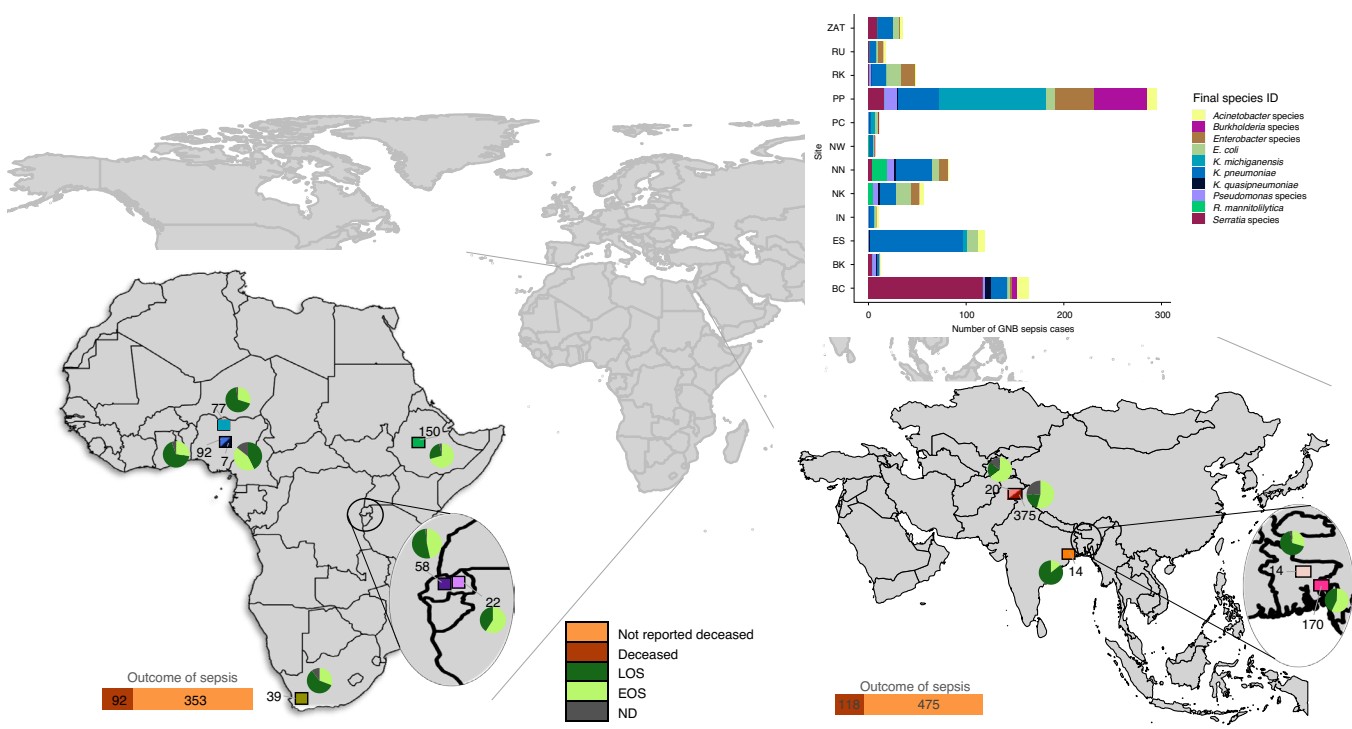

**Fig. 2 | World map showing the BARNARDS study clinical site locations.** The study sites are indicated by coloured squares. The African sites were located in Ethiopia (ES (green)), Nigeria (NK (cyan), NN (light blue) and NW (dark blue)), Rwanda (RK (dark purple) and RU (light purple)) and South Africa (ZAT; olive). The Asian sites were located in Bangladesh (BC (dark pink) and BK (light pink)), India (IN (orange)) and Pakistan (PC (peach) and PP (burgundy)). The numbers next to each clinical site location represent the total number of GNB identified. Inset: the stacked bar graph shows the distribution of the top ten GNB species recovered from blood cultures at the local sites. The onset of neonatal sepsis (EOS, LOS or ND) for GNB per clinical site is represented as a pie chart. The outcome of neonatal sepsis is shown for each continent.

species (n = 36) and *Ralstonia mannitolilytica* (n = 21) (Fig. 2). Among the GNB characterized herein, 401 were *Klebsiella*, with six species identified: *K. pneumoniae*, *K. variicola*, *K. quasipneumoniae*, *K. aerogenes*, *K. oxytoca* and *K. michiganensis* (Supplementary Fig. 4). *Burkholderia cenocepacia*, *K. michiganensis*, *R. mannitolilytica* and *S. marcescens* were mainly isolated from samples obtained from single sites in Pakistan, Nigeria and Bangladesh (Fig. 2).

Overall, *R. mannitolilytica*, *K. michiganensis*, *Burkholderia* species and *Pseudomonas* species caused more cases of EOS than other species (Supplementary Table 2 and Supplementary Fig. 5). Of note, similar proportions of fatal sepsis cases were caused by GNB on each continent (92/353 cases in Africa (21%) and 118/475 cases in Asia (20%)). We found that *Burkholderia* species or *K. michiganensis* sepsis infections were the most likely to be fatal (Supplementary Table 2 and Supplementary Fig. 5). However, there was a large proportion of missing data from certain clinical sites (due to data collection and entry error; Fig. 1), which limited the strength of conclusions.

**Interspecies and intraspecies diversity across clinical sites.** To understand the extent of inter- and intraspecies diversity, we aimed to perform WGS on all GNB. Multilocus sequence typing (MLST) was used primarily as an unambiguous tool to examine bacterial relatedness. As MLST can be performed easily via PCR of housekeeping genes, in addition to in silico MLST via WGS, assessing relatedness via this tool (where applicable) provides a dataset that can be extrapolated to existing data on neonatal sepsis where WGS may not be available.

MLST revealed substantial intraspecies diversity, and 40 previously unknown STs were assigned in 12 species (Table 1 and Supplementary Table 2). Fourteen STs were assigned in the *Klebsiella* genera (Table 1), including all three *K. aerogenes* STs (all from

Africa; n = 2 from Nigeria and n = 1 from South Africa; ST194–196). *K. michiganensis* was mainly ST180 (from PP in Pakistan; Table 1). Such low ST diversity and close phylogenetic relatedness, as shown by the core genome phylogeny (Supplementary Fig. 6), warrant further investigation. Similarly, we noticed that a single, previously unknown, *B. cenocepacia* ST (ST1621), also from PP, was dominant (Table 1 and Supplementary Fig. 7). *B. cenocepacia* ST1621 from PP and the *S. marcescens* isolates from BC were indistinguishable during core genome analysis (Supplementary Figs. 7 and 8). All local-level clusters will be studied further.

*Enterobacter cloacae* complex isolates were identified belonging to *E. cloacae*, *E. hormaechei*, *E. kobei*, *E. asburiae* and *E. ludwigii*. In total, seven different *Enterobacter* species (n = 80) with 28 STs were identified. The majority of *Enterobacter* species were found in Pakistan (n = 39), Nigeria (n = 19) and Rwanda (n = 14) (Fig. 1 and Supplementary Fig. 9). ST171 was common across sites in Africa and Asia; however, ST346 was only detected in Rwanda, ST523 was only detected in Pakistan and ST850 was only detected in Nigeria. Within *Enterobacter* species, 13 STs were assigned (Table 1) to *E. cloacae*, *E. hormaechei* and *E. ludwigii* isolates.

*Acinetobacter* species were recovered from ten out of 12 clinical sites in both Africa and South Asia, and 38 out of 49 (78%) were *Acinetobacter baumannii*. Of these, 17 out of 38 (45%) belonged to international clones (Pasteur MLST) ST1 and ST2 (Supplementary Tables 1 and 2).

**AMR of pathogens causing neonatal sepsis.** One aim of the BARNARDS study was to describe the AMR profiles of pathogens causing neonatal sepsis. For this, we performed agar dilution to determine the minimum inhibitory concentrations (MICs) of 19 antibiotics, including the current recommended first-line empirical

**Table 1 | STs for the most commonly identified species with a recognized MLST scheme**

| Species | Isolates characterized by WGS[a] | Number identified per clinical site[b] | Number of STs found | Prominent STs | STs predominantly found in specific clinical sites | Previously unknown STs |
|---|---|---|---|---|---|---|
| *A. baumannii* complex | 41 (38 ABA and 3 ANO) | BC (*n*=8), BK (*n*=2), ES (*n*=7), IN (*n*=3), NK (*n*=3), PC (*n*=1), PP (*n*=8), RK (*n*=2), RU (*n*=3) and ZAT (*n*=3) | 15 | ST1, ST2, ST575 and ST1106 | ST2 (BC) and ST1106 (PP) | ST1326 and ST1327 |
| *B. cenocepacia* | 56 | BC (*n*=1), PC (*n*=1) and PP (*n*=54) | 5 | ST1621 | ST1621 (PP) | ST1621 and ST1623 |
| *E. coli* | 75 | BC (*n*=3), ES (*n*=11), IN (*n*=2), NK (*n*=15), NN (*n*=7), NW (*n*=1), PC (*n*=3), PP (*n*=10), RK (*n*=15), RU (*n*=2) and ZAT (*n*=6) | 37 | ST10, ST69, ST131, ST410 and ST517 | ST410 (PC) and ST517 (RK) | ND |
| *E. cloacae* complex | 78 (1 EAS, 57 ECL, 18 EHO, 1 EKO and 1 ELU) | BC (*n*=2), NK (*n*=7), NN (*n*=9), NW (*n*=1), PP (*n*=39), RK (*n*=14), RU (*n*=5) and ZAT (*n*=1) | 34 | ST84, ST93, ST171, ST346, ST523 and ST980 | ST84 (PP), ST93 (PP), ST171 (PP), ST346 (RK), ST523 (PP) and ST980 (RK) | ST1236 and ST1238–ST1248 |
| *K. michiganensis/ K. oxytoca* | 122 (117 KMI and 5 KOX) | ES (*n*=5), PC (*n*=4), PP (*n*=111) and RK (*n*=2) | 5 | ST180 | ST180 (PP) | ST268 (KMI) and ST243–ST244 (KOX) |
| *K. pneumoniae* | 258 | BC (*n*=17), BK (*n*=2), ES (*n*=95), IN (*n*=5), NK (*n*=16), NN (*n*=37), NW (*n*=4), PC (*n*=2), PP (*n*=42), RK (*n*=15), RU (*n*=7) and ZAT (*n*=16) | 57 | ST15, ST35, ST37, ST39, ST218, ST307, ST348, ST443, ST464 and ST985 | ST15 (PP), ST35 (ES), ST37 (ES), ST218 (ES), ST307 (RK), ST442 (NN), ST464 (NN) and ST985 (ES) | ST4008, ST4410 and ST4411 |
| *K. quasipneumoniae* | 13 | BC (*n*=6), BK (*n*=1), ES (*n*=1), NK (*n*=2), NN (*n*=2) and PP (*n*=1) | 10 | ST4405 | ST4405 (BC) | ST4405–ST4407 and ST4409 |
| *K. variicola* | 5 | BC (*n*=1), PP (*n*=1), RK (*n*=2) and RU (*n*=1) | 5 | ND | ND | ST4404 and ST4412–ST4414 |
| *P. aeruginosa* | 23 | BC (*n*=1), BK (*n*=4), IN (*n*=1), NK (*n*=2), NN (*n*=3) PC (*n*=1), PP (*n*=9) and RK (*n*=2) | 14 | ST3235, ST1285 and ST3311 | ST1285 (BK) and ST3311 (PP) | ST3311 |
| *Salmonella enterica* | 7 | NK (*n*=6) and NN (*n*=1) | 4 | ST313 | ST313 (NK) | ND |

[a]Numbers of isolates characterized. For complexes, a breakdown by species is given. [b]Numbers of isolates from each clinical site. ABA, *A. baumannii*; ANO, *A. nosocomialis*; EAS, *Enterobacter asburiae*; ECL, *E. cloacae*; EHO, *Enterobacter hormaechei*; EKO, *Enterobacter kobei*; ELU, *Enterobacter ludwigii*; KMI, *K. michiganensis*; KOX, *K. oxytoca*.

treatments for neonatal sepsis, as well as carbapenems (the incidence of carbapenem-resistant GNB is increasing at an alarming rate), on 885 GNB (31 isolates were not recovered following storage at −80 °C after genomic DNA extraction for WGS; Supplementary Table 2 lists the isolates recoverable for WGS and/or MIC testing). Current data in LMICs focus on profiling AMR within certain species only, and are therefore not exhaustive across all pathogens causing sepsis.

GNB (*n*=885) were resistant to ampicillin (95%), cefotaxime (83%) and ceftriaxone (80%), whereas they were sensitive to meropenem (13%), imipenem (15%) and tigecycline (16%) (Fig. 3a). The MICs required to inhibit the growth of 50% of organisms (that is, $MIC_{50}$ values) of piperacillin/tazobactam, carbapenems (imipenem, meropenem and ertapenem), amikacin, fosfomycin, quinolones (ciprofloxacin and levofloxacin) and colistin were lower than their resistance breakpoints. The $MIC_{90}$ values of all antibiotics tested were higher than their resistance breakpoints, with the exception of tigecycline, for which the $MIC_{90}$ was lower than the epidemiological cut-off value for *Providencia* and *Proteus* species[20,21].

Overall, GNB isolates resistant to at least one of the cephalosporins tested were less likely to cause LOS than EOS (*P*=0.017; odds ratio (OR)=0.63; 95% confidence interval (CI)=0.43–0.92; Supplementary Table 3). For the statistical analysis, the outcome measurement was MFBS (deceased as response, alive as reference). Concomitant resistance to the three cephalosporins tested versus isolate susceptibility to all produced an odds ratio of 0.626; 95% CI 0.426–0.918; *P* = 0.017. In this way, concomitant resistance to the three cephalosporins tested among GNB isolates was less likely among infants who stayed alive compared to those who were deceased.

As a marker of extended-spectrum β-lactamase antibiotic resistance gene (ARG), *bla*$_{CTX-M-15}$ was inspected. It was detected in at least nine species (*n*=523 isolates) and found in isolates from all study sites. We also screened genomes for genes coding for carbapenem resistance (*bla*$_{NDM}$, *bla*$_{OXA-48}$-like variants and *bla*$_{KPC}$). There were 146 single carbapenemase genes in ten species (*n*=128 isolates), and two carbapenem resistance gene homologues were present in 24 isolates. *bla*$_{NDM-1}$ (*n*=90; Bangladesh, *n*=23; India, *n*=6;

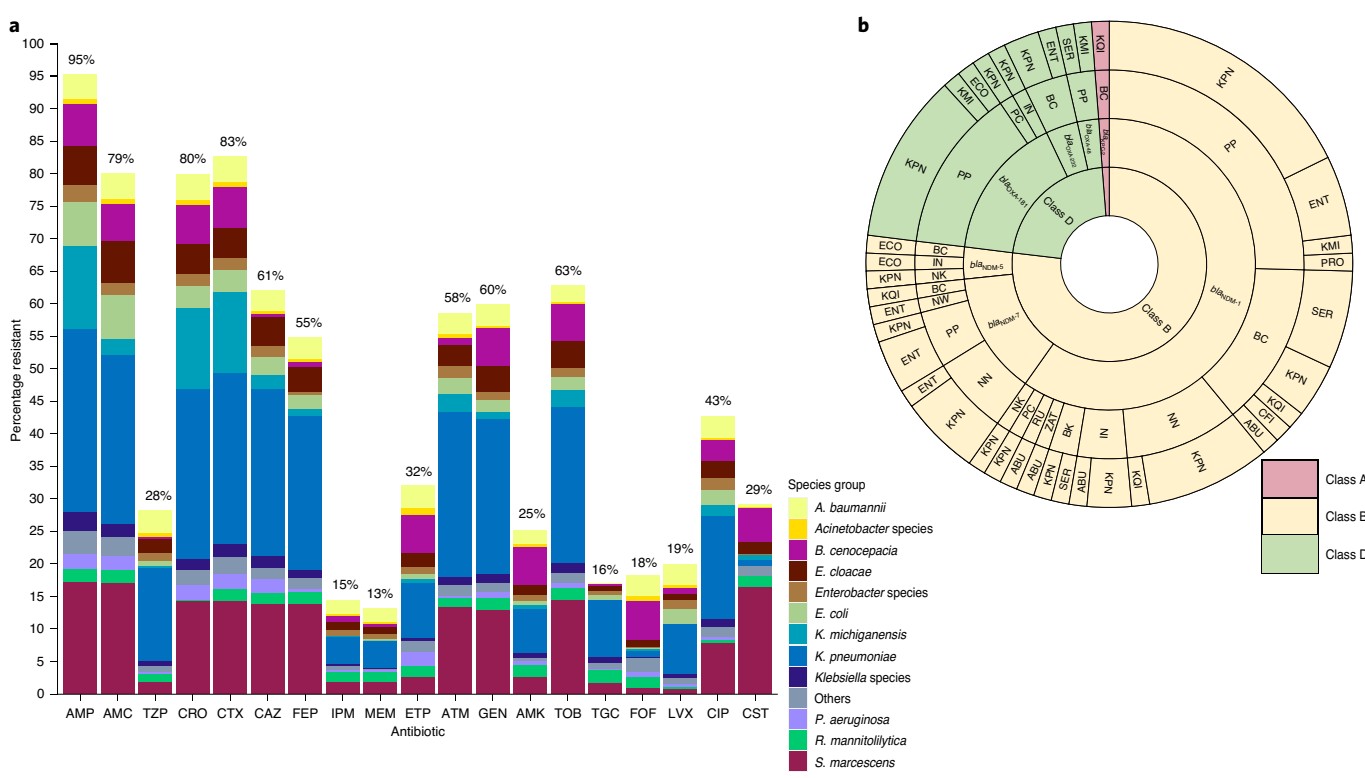

**Fig. 3 | AMR of neonatal sepsis-causing pathogens. a**, Percentages of antimicrobial-resistant aetiological agents of neonatal sepsis, coloured according to bacterial species/group ($n = 885$ isolates of GNB). The MICs of the antibiotics were determined by agar dilution and the results were interpreted according to EUCAST guidelines and documents[20,21]. AMC, amoxicillin/clavulanate; AMK, amikacin; AMP, ampicillin; ATM, aztreonam; CAZ, ceftazidime; CIP, ciprofloxacin; CRO, ceftriaxone; CST, colistin; CTX, cefotaxime; ETP, ertapenem; FEP, cefepime; FOF, fosfomycin; GEN, gentamicin; IPM, imipenem; LVX, levofloxacin; MEM, meropenem; TGC, tigecycline; TOB, tobramycin; TZP, piperacillin/tazobactam. **b**, Sunburst diagram detailing the class A (red), B (yellow) and D (green) carbapenemase resistance genes detected. The second ring from the centre shows the carbapenemase genes identified. The distributions across species and clinical sites are shown in the outer rings. ABU, *Acinetobacter baumannii*; CFI, *Citrobacter freundii*; ECO, *Escherichia coli*; ENT, *Enterobacter cloacae* complex; KMI, *Klebsiella michiganensis*; KPN, *Klebsiella pneumoniae*; KQI, *Klebsiella quasipneumoniae*; PRO, *Providencia rettgeri*; SER, *Serratia marcescens*.

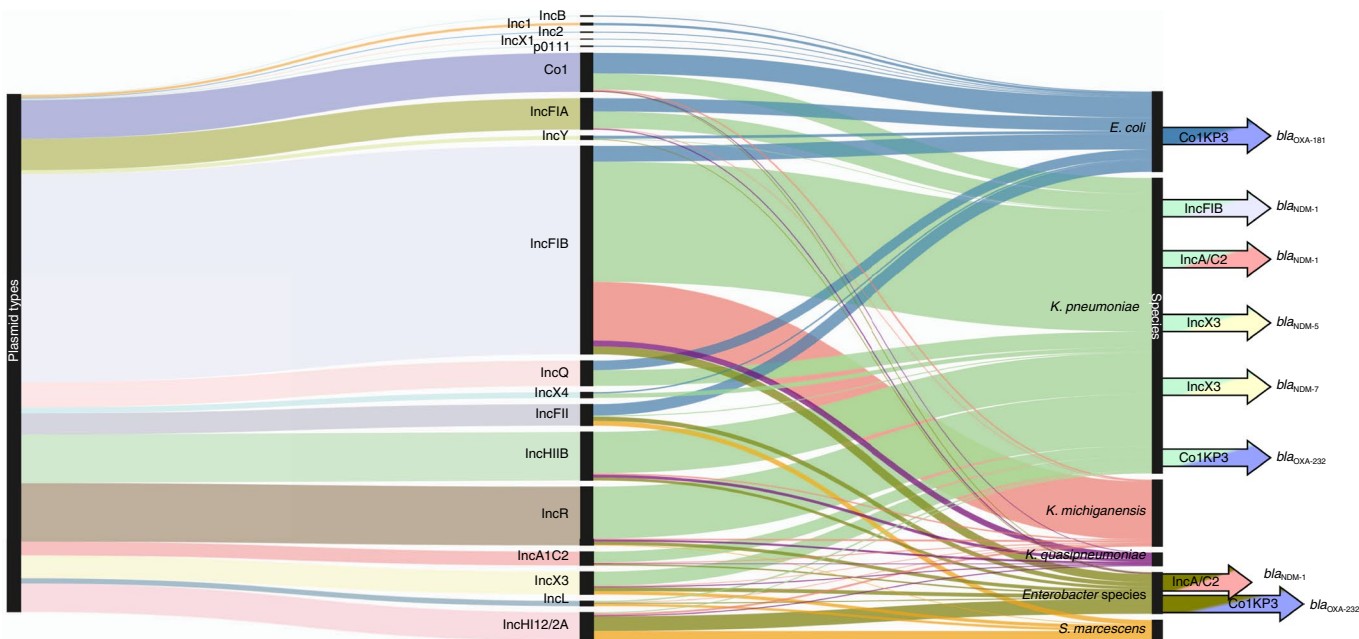

**Fig. 4 | Sankey diagram showing the distribution of the different plasmid types detected linking to the GNB in which they were found.** Plasmid types (left) found to carry carbapenemase AMR genes are colour coded and linked to the GNB species (right) in which the plasmid type was identified. Particular carbapenemase genes are shown on the far right.

Nigeria, $n = 16$; Pakistan, $n = 43$; Rwanda, $n = 1$; South Africa, $n = 1$) and $bla_{NDM-5}$ ($n = 3$; Bangladesh, $n = 1$; India, $n = 1$; Nigeria, $n = 1$) were mainly detected in clinical sites in South Asia, whereas $bla_{NDM-7}$ was predominantly recovered from Nigeria and Pakistan ($n = 19$; Bangladesh, $n = 1$; Nigeria, $n = 11$; Pakistan, $n = 7$) (Fig. 3b). In 79 GNB, $bla_{CTX-M-15}$ plus $bla_{NDM}$ and/or a $bla_{OXA-48}$-like variant were found. In total, 30 GNB carried a variant of the $bla_{OXA-48}$-like family, with $bla_{OXA-181}$ being the most frequent (Pakistan, $n = 22$; India, $n = 1$; Fig. 3b). $bla_{OXA-232}$ was only found in Bangladesh ($n = 5$), and the two isolates carrying $bla_{OXA-48}$ (one *K. michiganensis* and one *S. marcescens*) were from PP. $bla_{VIM}$ was found in three *Pseudomonas aeruginosa* isolates (Bangladesh, $n = 2$ (BC, $n = 1$; BK, $n = 1$); India, $n = 1$). Both $bla_{VIM}$ variants from Bangladesh were $bla_{VIM-2}$, whereas the variant detected in India was $bla_{VIM-6}$.

In *Enterobacter* species, $bla_{CTX-M-15}$ was found in isolates recovered from both Africa and South Asia ($n = 19$; Africa, $n = 16$; South Asia, $n = 3$); however, *Enterobacter* containing carbapenemase genes ($n = 18$) were largely recovered from South Asia (South Asia, $n = 16$; Africa, $n = 2$). Five *Acinetobacter* (*A. baumannii*, $n = 3$; *Acinetobacter bereziniae*, $n = 1$; *Acinetobacter nosocomialis*, $n = 1$) were found to have $bla_{NDM-1}$ (Africa, $n = 2$; South Asia, $n = 3$) (Fig. 3b). Additionally, we found $bla_{OXA-23}$ within 20 *A. baumannii* isolates.

The total number of ARGs possessed by each bacterial isolate is shown in Supplementary Table 4. For each ARG increase among *E. cloacae*, we observed a 13.2% decrease in the likelihood of neonates having LOS ($P = 0.016$; 95% CI = 0.77–0.97). No other associations between ARG and onset were found (Supplementary Table 3).

Worryingly, we found that 529 (60%) of the GNB isolates tested were resistant to the first-line empirical treatment for neonatal sepsis (both ampicillin and gentamicin).

**Plasmids and carbapenemase genes.** As many ARGs are carried on mobile genetic elements such as plasmids, we searched for plasmid replicon types in isolates from the different geographical areas and, where possible, analysed linkages between plasmid type and the carriage of specific carbapenemase genes. We detected 1,124 plasmids with 45 *inc* gene variants, which we categorized into 18 plasmid groups. From these, 1,093 were found within *E. coli* ($n = 169$), *K. pneumoniae* ($n = 623$), *K. michiganensis* ($n = 142$), *K. quasipneumoniae* ($n = 28$), *Enterobacter* species ($n = 87$) and *S. marcescens* ($n = 44$) (Fig. 4). There were 12 plasmid types found within the seven *S. enterica* isolates, seven among *Citrobacter* species and five among *K. variicola* isolates.

Within the six dominant plasmid carriers (Fig. 4), the most frequently detected *inc* type was IncFIB, with 255 out of 440 hits within *K. pneumoniae* genomes. $bla_{NDM-1}$ was found in IncA/C2 plasmids from *K. pneumoniae* and *Enterobacter* species from PP, as well as in IncFIB plasmids from *K. pneumoniae* from India (Fig. 4 and Supplementary Fig. 10). We found IncX3 plasmids carrying $bla_{NDM-5}$ in *K. pneumoniae* from NK, Nigeria, and multiple $bla_{NDM-7}$ in *K. pneumoniae* and *Enterobacter* species from Nigeria and Pakistan (Fig. 4 and Supplementary Fig. 10). Col plasmid types were identified within 82 genomes. We found ColKP3 plasmids carrying $bla_{OXA-181}$ or $bla_{OXA-232}$ in isolates of three different species: *E. coli*, *K. pneumoniae* and *Enterobacter* species (Fig. 4 and Supplementary Fig. 10).

Our bioinformatics analysis relied on the interrogation of short-read sequencing data; therefore, it was not possible to analyse the genetic context to link carbapenemase genes and *inc* type

for all genomes. Instead, a representative genome of each species/ ST with the largest contig carrying the carbapenemase gene was chosen to maximize the analysis of other genetic material present, including the *inc* gene ($n = 9$; Supplementary Fig. 10). This analysis demonstrated cases where the same carbapenemase gene variant was detected in the same plasmid type across different GNB, suggesting that successful dissemination and acquisition within multiple species may be occurring. We also found cases where the same carbapenemase ARG was detected in multiple different plasmids, furthermore evidencing the spread of AMR.

**Characterization of *K. pneumoniae*.** *K. pneumoniae* is an important cause of neonatal sepsis in LMICs; however, there are few data analysing this species beyond antimicrobial susceptibility testing. Here, we have shown that *K. pneumoniae* was the most frequently identified GNB; therefore, the genomic diversity of this collection was scrutinized to contextualize these isolates, both within this study collection and within previously known collections[15,16,22,23]. *K. pneumoniae* ($n = 258$) was found at all clinical sites (Figs. 1 and 5, Supplementary Table 2 and Supplementary Fig. 4)—predominantly, Ethiopia ($n = 95$), Nigeria ($n = 57$) and Pakistan ($n = 44$).

Genomics analysis within both the global[15,16,22] (Fig. 5a; see Supplementary Table 5 for literature search inclusion criteria) and neonatal sepsis context[23] (Fig. 5b) revealed high diversity of *K. pneumoniae*, with 156 STs from 17 countries spanning five continents. BARNARDS isolates clustered with previously reported neonatal isolates, including ST45, ST48 and ST348 (refs. [15,22,23]), but we also revealed distinct and new genetic lineages. The major AMR-related *K. pneumoniae* clades in Asia (ST11) and Europe (ST147 and ST307)[24] were also identified in this study. We only found ST307 in Rwanda ($n = 6$) and Nigeria ($n = 2$). However, ST258, a North American clade frequently associated with $bla_{KPC}$[25,26], was absent, which accords with the absence of $bla_{KPC}$ in this study. We did, however, detect one $bla_{KPC-2}$ gene in a *K. quasipneumoniae* from Bangladesh (Supplementary Fig. 10). While other studies have suggested that $bla_{KPC}$ *K. pneumoniae* causes neonatal sepsis, especially during nosocomial outbreaks[25], there is currently little evidence from countries in Africa or South Asia.

BARNARDS' *K. pneumoniae* were disseminated throughout the global phylogeny, with 57 STs (Table 1 and Fig. 5a). ST35 and ST37 were predominantly found in Ethiopia ($n = 38/39$ and $n = 29/30$, respectively). We found four ST35 *K. pneumoniae* from other neonatal sepsis publications; however, these sit on a distinct branch in the core genome phylogeny (Fig. 5b) and were more closely related to the single ST35 isolated from RU, Rwanda. ST15 isolates were largely isolated from Pakistan and all carried both $bla_{NDM-1}$ and $bla_{OXA-181}$ ($n = 22/27$; Fig. 5b). ST15 was almost exclusively found at the South Asian clinical sites, with a single ST15 found at NN, Nigeria. ST442 ($n = 6$) and ST464 ($n = 8$) were only found in NN and all isolates contained either $bla_{NDM-1}$ (ST442) or $bla_{NDM-7}$ (ST464).

Multiple different capsule types ($n = 47$ KL loci and $n = 12$ O loci) were identified in silico. ST15 isolates in Pakistan ($n = 23$) and India ($n = 1$) were all the O1v1:KL112 serotype, whereas single ST15 isolates in Bangladesh and Nigeria had different serotype combinations of O3b:KL38 and O1v1:KL48, respectively. Similarly, the ST35 isolates from Ethiopia were all O1v2:KL108, whereas *K. pneumoniae* from RU in Rwanda were O2v1:KL113. Of the eight ST348 isolates, of which seven were from South Africa and one was from

**Fig. 5 | Core genome characterization of *K. pneumoniae* isolates. a**, Five-hundred-and-fifty-nine isolates incorporating a global collection[23]. Blue shading indicates *K. pneumoniae* isolates from the BARNARDS collection. The branch labels are coloured according to country of origin. **b**, Detailed core genome characterization of 309 *K. pneumoniae* isolates ($n = 258$ BARNARDS). Yellow shading indicates isolates from other studies[15,22] causing neonatal sepsis. The outermost rings represent infant outcome (orange) and onset of sepsis (green), followed by the ST, where asterisks represent previously unknown STs. The leaf labels are the code names (coloured according to the study site) of isolates. The branch symbols in the centre denote the carriage of carbapenemase ARGs ($bla_{NDM}$ variants (circles) and $bla_{OXA-48}$ group variants (squares)). NA, not applicable.

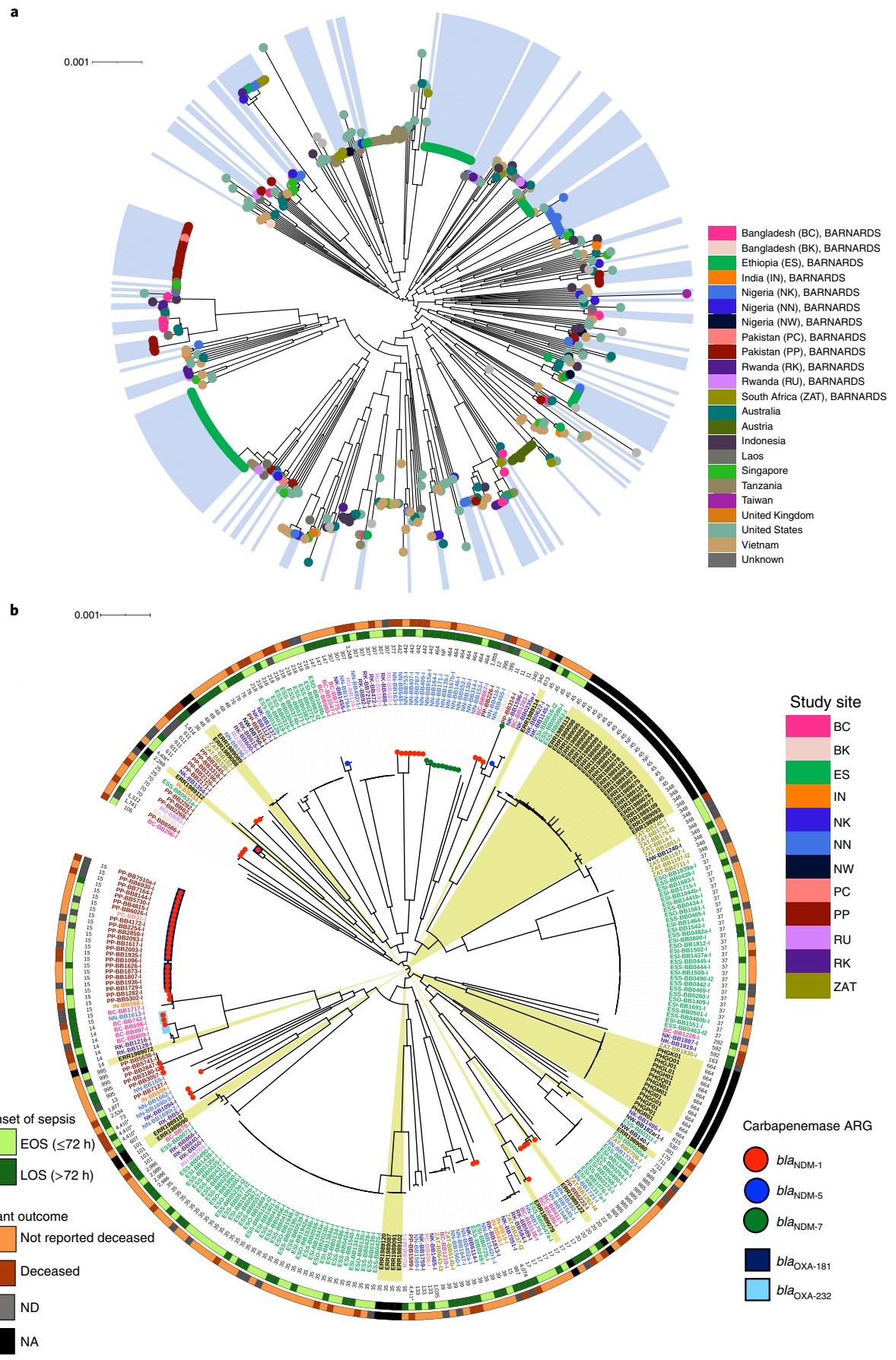

NW (Nigeria), all were O1v1:KL62 and all contained the yersiniabactin virulence gene.

In total, 115 isolates had a virulence score of 1, 3 or 4, indicating the presence of yersiniabactin and/or aerobactin/salmochelin virulence genes (see the Kleborate repository[27] and Source Data Fig. 5). The odds of LOS were 89% lower for infants with sepsis due to *K. pneumoniae* who had a virulence score of 3 or 4 compared with those with a score of 0 ($P = 0.04$; 95% CI = 0.014–0.90). Additionally, the odds of the outcome deceased were 14 times higher for infants with sepsis due to *K. pneumoniae* who had a virulence score of 3 or 4 compared with those with a virulence score of 0 ($P = 0.001$; 95% CI = 2.76–68.77; Supplementary Table 3). These results suggest that these genes may be involved in quicker onset and MFBS. Alternatively, they may reflect transmission of distinct isolates from the mother's microbiota[6] and from the clinical environment.

*K. pneumoniae* harboured multiple β-lactamase genes (Source Data Fig. 5). $bla_{CTX-M-15}$ was found in 220 out of 258 isolates across diverse STs and at all clinical sites, representing 42% (220/523) of total $bla_{CTX-M-15}$-positive GNB. Over one-quarter (26%; 69/258) harboured a variant of $bla_{NDM}$ (Figs. 4b and 5b). $bla_{NDM-1}$ ($n = 15$), $bla_{NDM-5}$ ($n = 1$) and $bla_{NDM-7}$ ($n = 9$) were found in Nigeria (mainly NN), whereas $bla_{NDM-1}$ was the dominant variant in South Asia (Fig. 4b).

All *K. pneumoniae* were resistant to ampicillin, cefotaxime, ceftriaxone and ceftazidime. *K. pneumoniae* concomitantly resistant to the three cephalosporins tested ($n = 255$) were significantly more likely to cause EOS than LOS ($P = 0.045$; OR = 0.41; 95% CI = 0.17–0.98; Supplementary Table 3). Also, 144 isolates had >15 ARGs. For each additional ARG, we observed a 5.7% ($P = 0.028$; 95% CI = 0.89–0.99) decrease in the odds of neonates having LOS compared to EOS.

Collectively, this analysis reveals a large degree of intraspecies diversity within *K. pneumoniae* pathogens causing neonatal sepsis in LMICs. While certain ST groups previously shown to be dominant in particular geographic regions were found during the BARNARDS study, we also detected several different ST groups carrying different carbapenemase ARGs and virulence determinants.

**Characterization of *E. coli*.** *E. coli* has previously been reported as a dominant GNB cause of neonatal sepsis across many different LMICs[18]. In light of this, we aimed to further characterize the BARNARDS *E. coli* isolates and to compare the results with existing WGS datasets (Supplementary Table 5 displays the literature search inclusion criteria). The 2014 enterotoxigenic *E. coli* collection[28] displayed large diversity and, when analysed with the BARNARDS dataset, 90 STs across four continents and 21 countries were detected (Fig. 6a). The *E. coli* analysed herein fell within four main clades of the extended phylogeny. The greatest numbers of *E. coli* in the BARNARDS study ($n = 75$) were in Nigeria ($n = 23$; 31%) and Rwanda ($n = 17$; 23%), although *E. coli* neonatal sepsis was identified in 11 of the 12 clinical sites (excluding BK, Bangladesh). A phylogenetic analysis of BARNARDS *E. coli* revealed four main groups, each containing multiple clades (Fig. 6b), with 37 STs detected. ST10 ($n = 9$), ST131 ($n = 6$), ST410 ($n = 5$) and ST69 ($n = 4$) were the most common. In the Mentzer et al.[28] collection, ST10 was found in five countries across Africa, Asia and South America. Of the nine ST10 *E. coli* characterized here, seven were from clinical sites within Africa (Ethiopia, $n = 1$; Nigeria, $n = 5$; Rwanda, $n = 1$) and two were from PP, Pakistan. Although all ST10 *E. coli* belonged

to the same phylotype (that is, group A), each isolate had a different O:H serotype profile and the phylogenetic tree (Fig. 6b) shows variability in the branch length, indicating genomic diversity within this ST group. Generally, we found highly variable O:H serotype classification ($n = 57$ O:H combinations), irrespective of the ST or phylogroup (A–F) (Source Data Fig. 6). Of the phylotypes, A ($n = 22$), B1 ($n = 15$) and B2 ($n = 18$) were the most common.

β-lactam and aminoglycoside ARGs were most commonly found in the South Asian isolates, with the exception of $bla_{CTX-M-15}$, which was also detected in *E. coli* from Africa. In total, 21 *E. coli* harboured $bla_{CTX-M-15}$ and belonged to a variety of STs, including ST131, ST405, ST410, ST10 and ST167. Carbapenemase genes were detected in only three out of 75 *E. coli*. The three isolates were from South Asia, and all concomitantly carried $bla_{CTX-M-15}$ plus $bla_{NDM-5}$ (ST167; $n = 2$) or $bla_{OXA-181}$ (ST410; $n = 1$).

No significant associations were found between phenotypic or genotypic AMR-related traits of *E. coli* and the clinical data assessed herein (Supplementary Table 3).

*E. coli* isolates were extremely diverse, suggesting that there are probably several important and worrisome lineages.

## Discussion

In the BARNARDS study, we established a methodological framework to capture and extensively characterize GNB species causing neonatal sepsis in LMICs. We isolated 916 isolates of GNB, characterized them to species level, used WGS to probe genome composition and MLST to assess intra- and interspecies diversity, and documented extremely high rates of AMR.

Most of the Gram-negative isolates from neonates with sepsis were resistant to at least one β-lactam and one aminoglycoside (597/885; 67%), as has been reported previously for cohorts in India[29] and 26 countries in Africa[18]. World Health Organization guidelines[30] stipulate ampicillin plus gentamicin as the first line of empirical treatment for neonatal sepsis and third-generation cephalosporins as the second line of treatment. Of note, many of the blood culture isolates from our study were resistant to both lines of treatment, meaning that treatment options are unlikely to be curative.

The identification of 58 different GNB species suggests that the aetiology of neonatal sepsis is complex. We report multiple different lineages causing infection within single species, many of which carry either resistant or putatively virulent mechanisms and several of which have previously been shown to cause neonatal sepsis (for example, ST35 and ST37 *K. pneumoniae*)[23,31]. The identification of high-risk clones, such as ST15 in *K. pneumoniae*[32] and the global clones ST1 and ST2 in *A. baumannii*, which are notorious for nosocomial infection[33], indicates the spread and persistence of problematic lineages in LMICs. In addition, through our comprehensive analysis, we identified 40 previously unknown STs, suggesting that well-known and previously unidentified lineages/ST groups are both co-existing and evolving.

A limitation of our study was the inability to follow up all neonates to 60 d (necessitating the exclusion of neonates who were lost to follow-up), which impacted our outcome data (Fig. 1). It is likely that additional local factors (such as the management of sepsis) contributed to mortality; therefore, we cannot attribute MFBS singularly to the presence/absence of genomic traits. Our statistical analyses were exploratory and should be interpreted as hypothesis generating only.

**Fig. 6 | Core genome characterization of *E. coli* isolates. a**, Three-hundred-and-sixty isolates incorporating a global collection[28]. Blue shading indicates *E. coli* isolates from the BARNARDS collection ($n = 75$). The branch labels are coloured according to country of origin. **b**, Detailed core genome characterization of 87 *E. coli* isolates ($n = 75$ BARNARDS). Yellow shading represents isolates from other studies[16,22] causing neonatal sepsis. The colours on the right represent infant outcome (orange) and onset of sepsis (green), followed by the ST. The numbers and code names (coloured according to study site) of isolates are also given. The branch symbols denote the carriage of carbapenemase ARGs ($bla_{NDM-5}$ (circles) and $bla_{OXA-181}$ (squares). NF, not found.

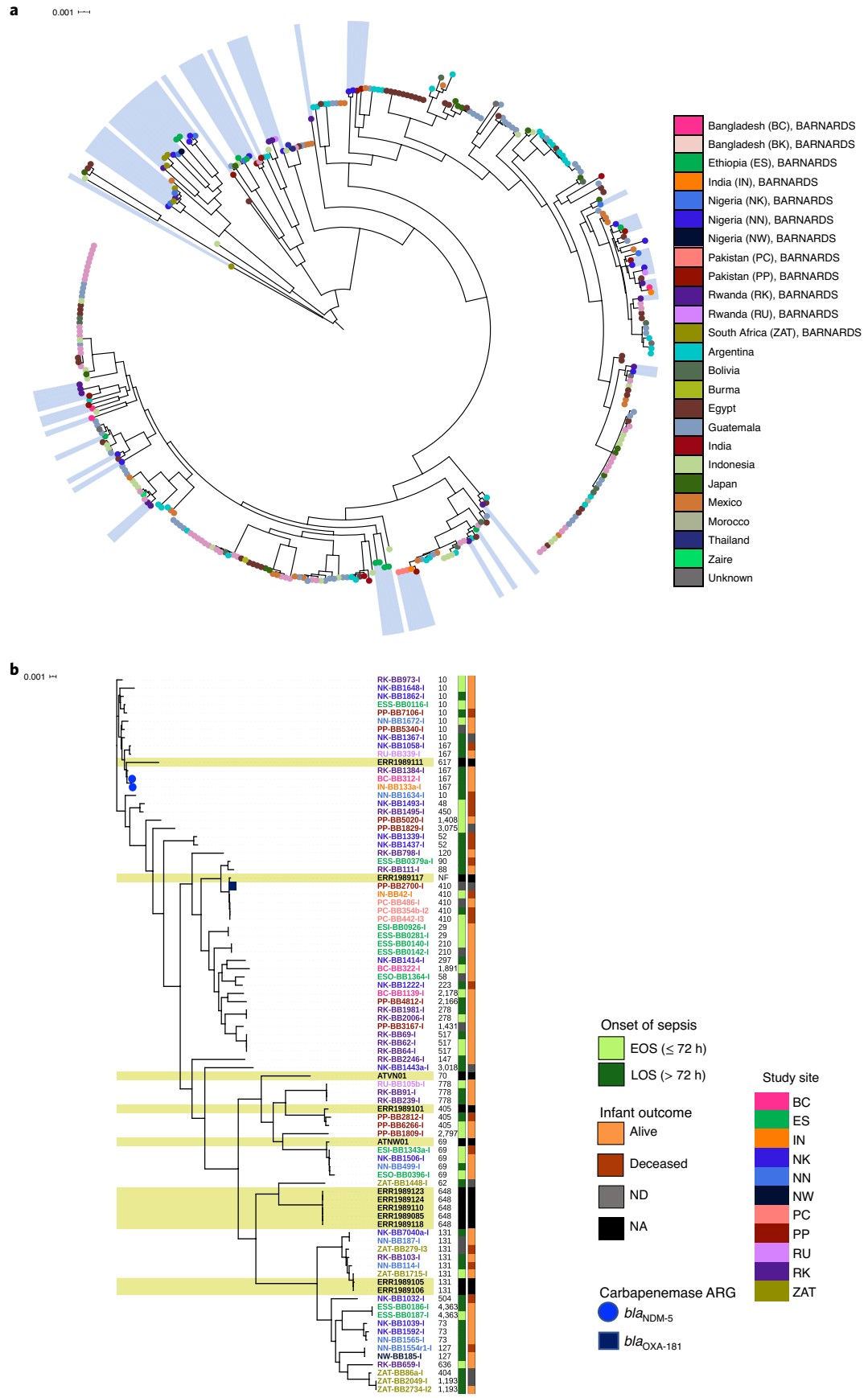

In summary, *Klebsiella*, *E. coli* and *Enterobacter* were the main GNB species responsible for sepsis in neonates. We report that 54% of isolated bacteria were resistant to at least one antibiotic within four to six classes of antibiotics, and observed widespread carriage of both resistance genes and virulence factors in GNB causing neonatal sepsis in LMICs. This large, observational study will inform

future research into effective antimicrobial therapies for neonatal sepsis, and may underpin improved infection control practices and could be useful in the development of vaccines for neonatal sepsis in LMICs.

## Methods

**Study design and processing of blood cultures at clinical sites.** A prospective cohort study was conducted through the BARNARDS network consisting of 12 clinical sites in seven countries in Africa and South Asia (Chattogram Maa-O-Shishu Hospital, Chattogram (BC) and Kumudini Women's Medical College, Mirzapur (BK) in Bangladesh; St. Paul's Hospital Millennium Medical College, Addis Ababa (ES) in Ethiopia; the Division of Bacteriology, ICMR-National Institute of Cholera and Enteric Diseases, Kolkata (IN) in India; National Hospital Abuja, Abuja (NN), Wuse District Hospital, Abuja (NW) and Murtala Muhammad Specialist Hospital, Kano (NK) in Nigeria; Pakistan Institute of Medical Sciences, Islamabad (PP) and Bhara Kahu Rural Health Centre, Bhara Kahu (PC) in Pakistan; University Central Hospital of Kigali, Kigali (RU) and Kabgayi Hospital, Kabgayi (RK) in Rwanda; and Tygerberg Hospital, Cape Town (ZAT) in South Africa. Ethical approval was obtained from the local ethics committee at each site before the start of the study (Supplementary Table 6). Between November 2015 and December 2017, women in labour or immediately postpartum were recruited prospectively following consent, and their neonates were followed up for the first 60 d of life (at 3, 7, 14, 28 and 60 d of life) or until study withdrawal/death. Neonates admitted to clinical sites showing signs of sepsis were also enrolled. The BARNARDS sample collection workflow is shown in Supplementary Fig. 11. Although it was not strictly a neonatal population, during this study, we employed the term neonate for all enrolments, including those between 30 and 60 d post-birth.

Documentation detailing the parameters for clinically diagnosing sepsis is available in Supplementary Fig. 1. The standard operating procedures for the laboratory processing of blood cultures and subsequent identification of bacteria, which were followed by all clinical sites (following agreement between clinical partners before the start of enrolment), are shown in Supplementary Fig. 2. Laboratory reagents (Liofilchem)—importantly, both agar media and antibiotic discs (used at the clinical sites)—were standardized throughout the network. Bacterial identification performed at each site was confirmed by WGS at Cardiff University. Antimicrobial susceptibility testing was performed twice: initially, at the local sites to guide treatment using antibiotic discs; and then at Cardiff University using the agar dilution method to establish the MICs (details below). The collected clinical data included onset of sepsis (EOS or LOS) and patient outcome following biological sepsis. Neonates that were lost to follow up were categorized along with neonates confirmed alive as 'not reported deceased'. For the purpose of this study, EOS and LOS were defined as sepsis occurring ≤72 h and >72 h after birth, respectively. If neonates showed clinical signs of sepsis at multiple time points within the first 60 d of life, additional blood cultures were analysed. All viable bacterial species were stored on charcoal swabs (Deltalab) for transport under UN3373 regulations to Cardiff University.

At Cardiff University, GNB isolates were plated onto chromogenic urinary tract infection media supplemented with vancomycin at 10 mg l$^{-1}$ (Liofilchem) and incubated aerobically overnight at 37 °C. Isolates were identified using a Microflex LT MALDI-TOF MS (Bruker Daltonik) with α-cyano-4-hydroxycinnamic acid matrix (Sigma–Aldrich). Bacterial isolates were stored in TS/72 beads (Technical Service Consultants) at −80 °C and the original swabs were stored at 4 °C. MICs were determined by agar dilution for a panel of 19 antibiotics and interpreted according to the EUCAST guidelines[20,21]. *E. coli* ATCC 25922 and *P. aeruginosa* ATCC 27853 strains were used as quality controls for GNB tests. Supplementary Table 7 depicts the panels of antibiotics and additional control strains used. Supplementary Table 8 defines the EUCAST interpretations used. The MIC$_{50}$ and MIC$_{90}$ values for each antibiotic was determined. The phenotypic metadata included the following AMR-related counts: carbapenem resistance (taken as resistance to ertapenem); methicillin resistance (taken as an oxacillin MIC > 2) and an AMR score (the number of antibiotics to which an isolate was resistant) (Supplementary Table 7).

**WGS.** A single bacterial colony was transferred into 1.8 ml LB broth and incubated at 37 °C and 180 r.p.m. for 18 h. Genomic DNA was extracted using a QIAamp DNA Mini Kit (Qiagen), with an additional RNAse step, on a QIAcube (Qiagen), and quantified using a Qubit Fluorometer 3.0. Genomic libraries were prepared using a Nextera XT V2 kit (Illumina) with bead-based normalization. A total of 48 isolates were multiplexed per sequencing run to provide a depth of coverage of >15×. Paired-end WGS was performed on an Illumina MiSeq using the V3 chemistry to generate fragment lengths of up to 300 base pairs (600 cycles).

**Bioinformatics analyses.** Bioinformatics analyses were performed using a high-performance computing cluster at Cardiff University (Advanced Research Computing at Cardiff (ARCCA)) and CLIMB (version 1.0)[34]. Paired-end reads (FASTQ) were subjected to quality control checks before downstream analysis. Trim Galore (version 0.4.3)[35] was used to remove the Nextera adapter sequences

and low-quality bases. Reports before and after read trimming were generated using FastQC (version 0.11.2)[36] and collated using MultiQC (version 1.7)[37]. The mean read length and number of sequences provided on the MultiQC reports were used to determine the sequencing coverage. Paired-end reads were overlapped using Flash (version 1.2.11)[38] and assembled into contigs using SPAdes (version 3.9.0)[39]. The trimmed FASTQ reads were mapped to the contigs using BWA (version 0.7.15)[40] and SAMtools (version 1.3.1)[41]. Pilon (version 1.22)[42] was used to assess any misassemblies/errors in base calling in the resulting mapped BAM file. Final genome assembly metrics were generated using QUAST (version 2.1)[43]. Bacterial species were identified using both BLAST nt (version 2.2.25; https://blast.ncbi.nlm.nih.gov/Blast.cgi; input = contigs)[44] and PathogenWatch (version 3.13.10; https://pathogen.watch; input = contigs). MLST, virulence and plasmid genomic profiles were characterized using SRST2 (version 0.2.0)[45] and the associated databases PlasmidFinder[46] and VFDB[47]. Genomes were screened for ARGs using ABRicate (version 0.9.7)[48] (databases NCBI[49] and ResFinder[50]).

Novel alleles and novel ST profiles were submitted to BIGSdb (version 1.25.1)[51]. The O:K locus profiles for all *Klebsiella* species were determined using Kaptive (version 0.7.0)[52] and Kleborate (version 0.2.0; https://github.com/katholt/Kleborate)[27]. The O:H serotype profiles for all *E. coli* isolates were determined using SerotypeFinder (version 2.0)[53], and SeqSero (version 1.0)[54] was used to determine serotypes for *Salmonella*. In silico *E. coli* phylotyping was performed using ClermonTyping (version 1.3.0)[55]. Genomes were annotated using Prokka (version 1.12)[30]. Strain relatedness analysis was performed using Roary (version 3.12.0)[56] to create a core genome alignment and FastTree (version 2.1.11) to generate maximum likelihood phylogenetic trees. Phylogenetic trees were mid-rooted, visualized and annotated using iTOL (version 5.7)[57]. The plasmid Sankey diagram was generated using the networkD3 package in R version 3.6.2. The immediate genetic context around carbapenemase genes was performed aligning outputs from ResFinder and PlasmidFinder (in ABRicate) with Mobile Element Finder (version 1.0.1) hosted by the Center for Genomic Epidemiology (https://cge.cbs.dtu.dk/services/MobileElementFinder/). The GBK annotation file from Prokka was then analysed for image production in Geneious Prime version 2020.1.2.

**Statistical analyses.** Statistical associations between clinical outcomes (onset of sepsis (EOS/LOS) and MFBS (alive/deceased)) and phenotypical and genomic traits were explored using univariable logistic regression models with the Wald test in SPSS version 26. The outcomes for the analyses regarding sepsis onset were EOS and LOS and those for MFBS were alive and deceased. Depending on whether predictor variables were continuous (AMR or ARG) or categorical (resistance versus non-resistance to ampicillin and gentamicin, concomitant resistance versus non-concomitant resistance to the three cephalosporins tested, resistance to at least one of the three cephalosporins tested versus to none, resistance versus non-resistance to ertapenem (a marker for carbapenems resistance) or *Klebsiella* species virulence scores), they were treated as covariates or factors, respectively. For species group analyses, only groups with $n = \geq 50$ isolates were included, except for *B. cenocepacia* isolates, which did not carry ARGs. Statistical significance was taken at $P \leq 0.05$, and estimated ORs are presented along with 95% CIs.

**Reporting Summary.** Further information on research design is available in the Nature Research Reporting Summary linked to this article.

## Data availability

Sequence reads have been submitted to the European Nucleotide Archive under project number PRJEB33565. Individual accession numbers and additional genomics data can be accessed in Supplementary Table 4 and the source data. The databases used for this study included VFDB (http://www.mgc.ac.cn/VFs/download.htm), NCBI (https://github.com/tseemann/abricate/tree/master/db/ncbi), ResFinder (https://github.com/tseemann/abricate/tree/master/db/resfinder), PlasmidFinder (https://bitbucket.org/genomicepidemiology/plasmidfinder/src/master), MLST (https://github.com/tseemann/mlst/tree/master/db/pubmlst), MGE (https://bitbucket.org/mhkj/mge_finder/src/master/me_finder/), SerotypeFinder (https://bitbucket.org/genomicepidemiology/serotypefinder/src/master) and SeqSero (http://www.denglab.info/SeqSero). Previously published datasets downloaded from the European Nucleotide Archive repository and used for comparative genomics analysis have the identifiers PRJEB2111, PRJEB2581 and PRJEB20875. The following genomes were downloaded from NCBI: PHGE01000000–PHGR01000000, ATNW00000000 and ATNV00000000. Source data are provided with this paper.

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

## Acknowledgements

We acknowledge Liofilchem for continued support in the distribution of microbiology products to enable standard operating procedures across the clinical sites. We acknowledge J. Parkhill for advice and guidance regarding the phylogenetic analyses.

We acknowledge Wales Gene Park and ARCCA for continued bioinformatics support and infrastructure availability. Bioinformatics analysis was largely undertaken using the supercomputing facilities at Cardiff University, which were operated by Advanced Research Computing at Cardiff (ARCCA) on behalf of the Cardiff Supercomputing Facility and HPC Wales and Supercomputing Wales projects. We acknowledge support from Supercomputing Wales, which is partly funded by the European Regional Development Fund via the Welsh Government. We thank the team of curators for the databases hosted on PubMLST (https://pubmlst.org/databases/). We also thank the curators of the Institut Pasteur MLST and Whole-Genome MLST databases for curating the *Klebsiella* data and making them publicly available at http://bigsdb.pasteur.fr. We thank M. Islam for providing access to the clinical sites and epidemiology data in Bangladesh. We acknowledge R. Kamran, the microbiologist from the Padmashree Institute of Management and Sciences, who sadly passed away in 2018. We thank the team at the Bill & Melinda Gates Foundation; namely, P. Srikantiah, R. Izadnegahdar, K. Klugman and S. Vernam. The BARNARDS study was funded by two awards (US$4.28 million (OPP1119772) and US$849,000 (OP1191522)) from the Bill & Melinda Gates Foundation.

## Author contributions

K.S. and M.J.C. designed and guided the study, performed the analysis and wrote the manuscript. K.S., E.P. and J.M. performed the WGS experiments. K.S. and R.A. performed the bioinformatics analysis. M.J.C. contributed to the bioinformatics analysis. E.P. and K. Thomson contributed equally. K. Thomson performed the MIC experiments. K. Thomson and M.J.C. analysed the MIC dataset. K.S. and M.J.C. produced the figures. C.D., R.M., D.G., K. Taiyari and K.H. provided the finalized epidemiological and clinical dataset. M.J.C., D.G. and K. Taiyari performed the statistical analysis. K.S., M.J.C., E.P., K. Thomson, C.D., C.A., A.F., T.H., J.M. and M.N. performed the microbiology culture and sample processing at Cardiff University. G.J.C., D.B., S.S., S.B., P.C., S.M., K.I., F.M., S.U., R.Z., H.S., A.M., J.-B.M., A.R., L.G., S.M., A.N.H.B. and A.W. collected and processed the blood cultures and collected clinical data at the clinical sites. T.R.W., M.J.C., R.M., G.J.C., S.B., K.I., R.Z., J.-B.M. and S.M. designed the BARNARDS study.

## Competing interests

The authors declare no competing interests.

## Additional information

**Correspondence and requests for materials** should be addressed to K.S. or M.J.C.

## BARNARDS Group

**Kirsty Sands**[1,2,25], **Maria J. Carvalho**[1,3,25], **Edward Portal**[1], **Kathryn Thomson**[1], **Calie Dyer**[1,4], **Chinenye Akpulu**[1,5,6], **Robert Andrews**[1], **Ana Ferreira**[1], **David Gillespie**[4], **Thomas Hender**[1], **Kerenza Hood**[4], **Jordan Mathias**[1], **Rebecca Milton**[1,4], **Maria Nieto**[1], **Khadijeh Taiyari**[4], **Grace J. Chan**[7,8,9], **Delayehu Bekele**[9,10], **Semaria Solomon**[11], **Sulagna Basu**[12], **Pinaki Chattopadhyay**[13], **Suchandra Mukherjee**[13], **Kenneth Iregbu**[5], **Fatima Modibbo**[5,6], **Stella Uwaezuoke**[14], **Rabaab Zahra**[15], **Haider Shirazi**[16], **Adil Muhammad**[15], **Jean-Baptiste Mazarati**[17], **Aniceth Rucogoza**[17], **Lucie Gaju**[17], **Shaheen Mehtar**[18,19], **Andre N. H. Bulabula**[19,20], **Andrew Whitelaw**[21,22] and **Timothy R. Walsh**[1,24]

A full list of members and their affiliations appears in Supplementary Table 9.

# nature research

# Reporting Summary

Nature Research wishes to improve the reproducibility of the work that we publish. This form provides structure for consistency and transparency in reporting. For further information on Nature Research policies, see our Editorial Policies and the Editorial Policy Checklist.

## Statistics

For all statistical analyses, confirm that the following items are present in the figure legend, table legend, main text, or Methods section.

| n/a | Confirmed | |
|---|---|---|
| ☐ | ☒ | The exact sample size (*n*) for each experimental group/condition, given as a discrete number and unit of measurement |
| ☐ | ☐ | A statement on whether measurements were taken from distinct samples or whether the same sample was measured repeatedly |
| ☐ | ☒ | The statistical test(s) used AND whether they are one- or two-sided<br>*Only common tests should be described solely by name; describe more complex techniques in the Methods section.* |
| ☐ | ☐ | A description of all covariates tested |
| ☐ | ☐ | A description of any assumptions or corrections, such as tests of normality and adjustment for multiple comparisons |
| ☐ | ☒ | A full description of the statistical parameters including central tendency (e.g. means) or other basic estimates (e.g. regression coefficient) AND variation (e.g. standard deviation) or associated estimates of uncertainty (e.g. confidence intervals) |
| ☐ | ☒ | For null hypothesis testing, the test statistic (e.g. *F*, *t*, *r*) with confidence intervals, effect sizes, degrees of freedom and *P* value noted<br>*Give P values as exact values whenever suitable.* |
| ☐ | ☐ | For Bayesian analysis, information on the choice of priors and Markov chain Monte Carlo settings |
| ☐ | ☐ | For hierarchical and complex designs, identification of the appropriate level for tests and full reporting of outcomes |
| ☐ | ☐ | Estimates of effect sizes (e.g. Cohen's *d*, Pearson's *r*), indicating how they were calculated |

*Our web collection on statistics for biologists contains articles on many of the points above.*

## Software and code

Policy information about availability of computer code

| Data collection | No software was used in the data collection. At the low middle income countries, research nurses completed questionnaires with the women approaching labor. These questionnaires were either transcribed onto paper, due to availability of resources/infrastructure, i.e. Internet access, and later uploaded into Bristol Online survey (BOS) or directly entered into BOS using a tablet device provided by the project. |
|---|---|
| Data analysis | CLIMB (v1.0)<br>Trimgalore (v0.4.3)<br>fastqc (v0.11.2)<br>MultiQC (v1.7)<br>Flash (v1.2.11)<br> SPAdes (v3.9.0)<br>BWA (v.0.7.15)<br>samtools (v1.3.1)<br>Pilon (v1.22)<br>quast (v.2.1)<br>Blast nt (https://blast.ncbi.nlm.nih.gov/Blast.cgi) (v2.2.25)<br>PathogenWatch (v.3.13.10; https://pathogen.watch)<br>srst2 (v0.2.0)<br>ABRicate (v0.9.7)<br>BIGSbd (v1.25.1)<br>Kaptive (v0.7.0)<br>and Kleborate (v0.2.0)<br>SerotypeFinder (v2.0)<br>SeqSero (v1.0)<br>ClermonTyping (v.1.3.0) |

```
Prokka (v1.12)
Roary (v3.12.0)
FastTree (v2.1.11)
iTOL (v5.7)
networkD3 package, Rv3.6.2
Geneious prime (2020.1.2)
```

For manuscripts utilizing custom algorithms or software that are central to the research but not yet described in published literature, software must be made available to editors and reviewers. We strongly encourage code deposition in a community repository (e.g. GitHub). See the Nature Research guidelines for submitting code & software for further information.

## Data

Policy information about availability of data

All manuscripts must include a data availability statement. This statement should provide the following information, where applicable:
- Accession codes, unique identifiers, or web links for publicly available datasets
- A list of figures that have associated raw data
- A description of any restrictions on data availability

Sequences reads were submitted to the European Nucleotide Archive (ENA) and given the project number PRJEB33565. A list of individual accession numbers for 916 Gram-negative bacteria can be found in the source data (to accompany figure 4,5 & 6).

Databases used within this study:
VFDB: http://www.mgc.ac.cn/VFs/download.htm
NCBI: https://github.com/tseemann/abricate/tree/master/db/ncbi
Resfinder: https://github.com/tseemann/abricate/tree/master/db/resfinder
Plasmidfinder: https://bitbucket.org/genomicepidemiology/plasmidfinder/src/master
mlst: https://github.com/tseemann/mlst/tree/master/db/pubmlst
MGE: https://bitbucket.org/mhkj/mge_finder/src/master/me_finder/
Serotype finder: https://bitbucket.org/genomicepidemiology/serotypefinder/src/master
Seqsero: http://www.denglab.info/SeqSero

Previously published datasets downloaded from the ENA repository used for comparative genomics analysis: PRJEB2111, PRJEB2581 and PRJEB20875. Genomes were downloaded from NCBI: PHGE01000000-PHGR01000000, ATNW00000000, ATNV00000000.

# Field-specific reporting

Please select the one below that is the best fit for your research. If you are not sure, read the appropriate sections before making your selection.

☒ Life sciences　　　☐ Behavioural & social sciences　　　☐ Ecological, evolutionary & environmental sciences

For a reference copy of the document with all sections, see nature.com/documents/nr-reporting-summary-flat.pdf

# Life sciences study design

All studies must disclose on these points even when the disclosure is negative.

| | |
|---|---|
| Sample size | The sampling method was purposive and a formal sample size calculation was not conducted.<br><br>Based on previous studies led by PI Professor Timothy Walsh (unpublished studies/awaiting publication), BARNARDS anticipated the enrollment level between 500-2000 neonates per clinical site for the duration of the study (depending on geographical location i.e. smaller rural site would have a smaller catchment area). With the anticipated sepsis rate (based on limited public accessible data), this would capture between 1,000-2,500 bacterial (both Gram-negative and Gram-positive) blood culture positive sepsis cases across seasonal variation with an estimated 1,000 Gram-negative sepsis cases. |
| Data exclusions | The following exclusion criteria was pre-defined: the sepsis case infant/mother sampling pair was excluded in the case of a still born. Following this, data was retrospectively excluded based on the following criteria:<br>- Incomplete questionnaire; missing multiple data points in the epidemiological dataset<br>- Mother asked for infant withdrawal<br>- Error/substantial inconsistencies in the questionnaire - laboratory sampling match up. |
| Replication | N/A - Findings were not replicated as this manuscript is a full genomics characterisation of sepsis causing bacteria from low middle income countries (LMICs). All viable isolates eligible for sequencing were included into the analysis. |
| Randomization | Randomization was not relevant to the study. All women approaching labor were (following consent) enrolled onto the study. If the neonate/infant presented with sepsis, the infant was also enrolled to allow a blood culture to be taken. |
| Blinding | Blinding was not necessary for this study as we were characterizing all blood culture isolates along with the corresponding infant's clinical data. |

# Reporting for specific materials, systems and methods

We require information from authors about some types of materials, experimental systems and methods used in many studies. Here, indicate whether each material, system or method listed is relevant to your study. If you are not sure if a list item applies to your research, read the appropriate section before selecting a response.

## Materials & experimental systems

| n/a | Involved in the study |
|-----|------------------------|
| ☒ ☐ | Antibodies |
| ☒ ☐ | Eukaryotic cell lines |
| ☒ ☐ | Palaeontology and archaeology |
| ☒ ☐ | Animals and other organisms |
| ☐ ☒ | Human research participants |
| ☒ ☐ | Clinical data |
| ☒ ☐ | Dual use research of concern |

## Methods

| n/a | Involved in the study |
|-----|------------------------|
| ☒ ☐ | ChIP-seq |
| ☒ ☐ | Flow cytometry |
| ☒ ☐ | MRI-based neuroimaging |

## Human research participants

Policy information about studies involving human research participants

**Population characteristics**

BARNARDS was a multi-site international prospective observational study incorporating two recruitment pathways:
i.) Birth-Cohort: All mothers in labour admitted to clinical-sites were recruited prospectively and their infant(s) followed up until 60-days old or death.
ii.) Infant Admissions (IA): Infant(s) admitted to clinical-sites showing signs of suspected sepsis in the first 60-days of life until 60- days old or death.
2
nature research | reporting summary October 2018

For this study, isolates recovered from blood cultures were included into the characterisation of infant sepsis irrespective of cohort pathway.
General population characteristics of the mothers' (oustide of the scope of this manuscript): <10% previously had stillbirth, approx. 25% were first time mothers', 75% were aged between 21-35 years old. Infants' presenting with sepsis were followed up for 60 days of life. Onset of sepsis was recorded, early onset (EOS) <72h, and late onset (LOS) >72h.

**Recruitment**

BARNARDS recruited from 12 clinical sites from Rwanda, Bangladesh, Ethiopia, Nigeria, Pakistan, India and South Africa. Where possible, large public hospitals were chosen.

Following the presentation of information regarding the study, mothers in labour were enrolled into the study. Consent was collected by trained research staff and using local languages. Corresponding neonates presenting with clinical signs of sepsis were then enrolled into the study.
Additionally, neonates not born within the clinical sites that were admitted with clinical signs of sepsis were also enrolled into the study following consent from the mother. The corresponding mothers were also enrolled into the study for the collection of demographic data.
Neonatal follow-up was carried out at day 3, 7, 14, 28, and 60 by research nurses either face-to-face or by telephone. Neonates remained in the study until 60 days old, withdrawal, or death.
This study incorporated two recruitment pathways to include both neonates born within the clinical sites, and also neonates in the larger catchment areas presenting to the hospital with signs of sepsis.

**Ethics oversight**

Site committees Named PI Reference(s) Approval date(s)
BC - Ethical Review Committee, Bangladesh Institute of Child Health Samir Kumar Saha BICH-ERC-4/3/2015 15/09/2015
BK - Ethical Review Committee, Bangladesh Institute of Child Health Samir Kumar Saha BICH-ERC-4/3/2015 15/09/2015
ES - Boston Children's Hospital Grace Chan IRB-P00023058 11/08/2016
IN - Institutional Ethics Committee, National Institute of Cholera and Enteric Diseases and Institue of Post Graduate Medical Education and Research, IPGME&R Research Oversight Committee Sulagna Basu A-I/2016-IEC and Inst/IEC/2016/508 17/11/2016 and 04/11/2016
NK - Kano State Hospitals Management Board Kenneth Iregbu 8/10/1437AH 13/07/2016
NN - Health Research Ethics Committee (HREC), National Hospital, Abuja Kenneth Iregbu NHA/EC/017/2015 27/04/2015
NW - Health Research Ethics Committee (HREC), National Hospital, Abuja Kenneth Iregbu NHA/EC/017/2015 27/04/2015
PC - Shaheed Zulfiqar Ali Bhutto Medical University, Pakistan Institute of Medical Sciences (PIMS) Islamabad Rabaab Zahra NA, signed letter from Prof. Tabish Hazir 27/05/2015
PP - Shaheed Zulfiqar Ali Bhutto Medical University, Pakistan Institute of Medical Sciences (PIMS) Islamabad Rabaab Zahra NA, signed letter from Prof. Tabish Hazir 27/05/2015
RK - Republic of Rwanda, National Ethics Committee Jean-Baptiste Mazarati No342/RNEC/2015 10/11/2015
RU - Republic of Rwanda, National Ethics Committee Jean-Baptiste Mazarati No342/RNEC/2015 10/11/2015
ZAT - Stellenbosch University and Tygerberg Hospital, Research projects, Western Cape Government Shaheen Mehtar N15/07/063 04/12/2015 and 02/02/2016

Note that full information on the approval of the study protocol must also be provided in the manuscript.

