## [Peer Review File · Nature Microbiology]

Peer Review Information

Journal: Nature Microbiology

Manuscript Title: Characterization of antimicrobial resistant Gram-negative bacteria that cause neonatal sepsis in seven low and middle-income countries

Corresponding author name(s): Kirsty Sands

Reviewer Comments & Decisions:

Decision Letter, initial version:
--

Dear Kirsty,

Thank you for your continued patience while your manuscript "Genomic analysis of bacteria causing infant sepsis in low- and middle-income countries" was under peer review at Nature Microbiology. Please accept our apologies for the time it has taken us to contact you with a decision on your study, which was due to a delayed referee report. Unfortunately, we had to proceed without receiving the final missing report because even after multiple reminders, one referee did not provide the report. Your study has been seen by two referees, whose expertise and comments you will find at the end of this email. In the light of their advice, we have unfortunately decided that we cannot offer to publish your manuscript in Nature Microbiology.

From the reports, you will see that while the referees find your work of some potential interest and acknowledge the scale and the detailed analysis of your study, they raise serious concerns over the focus and level of insight of your study, the overall novelty of the findings, and the presentation of the results. Referee 1 feels that the study "lacks a real story or focus" and that the analysis "is nothing ground-breaking and the genomic analysis does not seem to add anything particularly new". Referee 2 states that "the description is so "high level" as to render it uninformative". These criticisms are important, and given the length of time that it would likely take to address them thoroughly and the uncertain outcome of such revision, we must decline publication of your work in Nature Microbiology at this stage and consider that your best option is to submit the manuscript in its current form to another journal.

However, we remain very interested in this manuscript. If you can restructure the manuscript to highlight more clearly the clinical and epidemiological inferences from these data, and make this analysis the main focus of the article, we may be able to reconsider. The novel, confirmatory and actionable outcomes from this analysis needs to be made clear as does how this body of evidence adds to existing knowledge of causes of infant sepsis.

Please note that procedurally, we would reconsider a revised version as an appeal to this decision. Please note that we would reassess novelty with respect to existing literature at the time of appeal and would be unlikely to contact our referees again unless we felt that their concerns had been satisfied in full. In the case of a successful appeal and eventual publication, the received date would be that of the revised manuscript. Our chief editor, Dr. Susan Jones, will be happy to discuss any questions you may have regarding a potential appeal and resubmission of a revised version.

I am sorry that we cannot be more positive on this occasion, but hope that you find the referees' comments helpful.

Reviewer Expertise:

Referee #1: {REDACTED}

Referee #2: {REDACTED}

Reviewers Comments:

Reviewer #1 (Remarks to the Author):

The paper "Genomic analysis of bacteria causing infant sepsis in low- and middle-income countries" is a description of the organisms isolates from the Gates funded BARNARDS (Burden of Antibiotic Resistance in Neonates in Developing Societies) study and is a comprehensive description of the organisms associated with sepsis in several sites. The study is excessively broad and lacks any real focus and whilst I appreciate the effort and the detailed analysis the paper represents a huge data dump with no real story or focus. Such multi-pathogen papers are hard to publish as these fields tend to be dominated by groups working on specific organisms and whilst the study is an excellent framework and whilst the analysis is good is nothing ground-breaking and the genomic analysis does not seem to add anything particularly new. There are no risk factors presented (as suggested) and no link with clinical data (as also suggested) or indeed any other variable that was likely collected and in its current state the paper lacks a message and adds little to these fields. There are some interesting insights into these organisms and a more detailed/specific focus on one of them would likely be a more interesting and novel addition to the field. In summary the paper, whilst a huge effort, does not really add anything new to our knowledge of these organisms (apart from scale). Without some additional clinical or epidemiological data, it is a collection of trees and some largely illegible diagrams of AMR genes. It is a real pity as a some form of focus and it could rival the GEMS study or other large Gates studies conducted in LMICs.

Reviewer #2 (Remarks to the Author):

In "Genomic analysis of bacteria causing infant sepsis in LMIC" Sands and colleagues describe an analysis of 1067 bacterial genomes from the Barnards consortium. The collection represents a major, recent global snapshot and the genomic analysis is well performed and well described. In that sense the work in this manuscript should be commended and recognised as important.

The manuscript is not without significant flaws. The description is so “high level” as to render it uninformative. This is evidenced by the fact that there is no concluding line to the abstract, which instead peters out after the results summary. I think there are several reasons for this. Firstly, the authors have simply situated several key genera within their collection in a global context, rather than placing them more fully in the clinical context in which they were isolated, that of neonatal sepsis in diverse LMICs. The authors rightly highlight that a strength of BARNARDS is that it goes from clinical observation through classical microbiology to genomics, yet all that is really on display here is the genomics, and there is so much diversity that none of the undoubtably important detail emerges. For example were specific lineages/virulence factors/AMR patterns associated with elevated mortality? Or specific outbreaks?

It feels to me as if the authors have got lost in trying to summarise this vast data set and consequently not pulled out any interesting narratives – consequently the discussion largely restates the results and does not discuss the data. They highlight a couple of potentially interesting stories i.e. regarding *K. michiganensis* and *Staph aureus* and then do not go into any further detail. I wonder if a better approach would be to submit one manuscript per major genus for the most common genera and then go into much greater depth. Instead of submitting 3-4 panel figures that are difficult to interpret even when one zooms in (perhaps the fault of the manuscript submission process?), the authors could then provide 3 large, individual high resolution figures per genus. They could then drill into genus and geography specific stories, comparing and contrasting differences within and between study sites and make much more of the clinical metadata associated with each isolate. I would have thought a *Klebsiella* paper on its own would be worthy of consideration by a Nature journal.

Lastly BARNARDS is supposed to have an AMR lens and indeed AMR genes are dutifully reported, however AMR is barely commented upon in the discussion. Again this theme suffers from being so high level for each genus.

There are various ways in which I think the manuscript needs extensive editing (see below), but I think its greatest weakness is that the authors try to compress too much data into one overarching manuscript and consequently fail to say anything especially interesting.

Introduction:

The manuscript reads as if written by a bioinformatician – the bioinformatic methods and results are well described, but there is a disconnect between this is and the epidemiology and broader clinical context. For example, the introduction is poorly written i.e. line 47 “Infectious diseases mostly affect children in LMC...” is weak – there is a greater burden of severe bacterial infections in LMIC and clinical outcomes are worse, but there is no shortage of infections in high income settings, indeed 50% of primary care consultations in the UK are still related to suspected infection.

Why have they chosen the term “infant sepsis” rather than “neonatal sepsis”? There may be a good reason and if so this should be specified. Certainly “neonatal sepsis” has more traction and would gain the interest of neonatologists. Looking at supplemental fig 1 it looks as if participants recruited up to 60 days so it is not a strict neonatal population?

Line 56-58 poorly written and should be re-phrased, not sure what the authors are trying to say. As I cannot understand the point made in this sentence, the next one is meaningless to me.

Line 60-62, the authors should reflect on and cite the recent Lancet ID systematic review on neonatal sepsis in SSA (Okomo et al 2019, PMID: 31522858), also the single centre data from Iroh-Tam et al CID 2018 (PMCID: PMC6579959)

Line 68, I dislike "uniquely" and generally claims of primacy should be discouraged

Methods

I would like to see a supplemental table with all the ethics committees listed and the relevant reference numbers or a reference to where these can be seen. Did Cardiff not provide ethics and thus sponsorship in addition to the local RECs?

Line 89-90 a lab SOP would not have provided an SOP for "the clinical diagnosis of sepsis" but for the laboratory processing of blood cultures and the subsequent identification of bacteria.

What were the criteria for "clinical diagnosis of sepsis" I appreciate this may appear elsewhere, but the criteria for recognising a case of neonatal sepsis and then obtaining blood for culture should be presented in slightly more detail at line 106.

Is the lancet ID paper (ref 16) in press? If so say so, if not I would be cautious about citing it and the manuscript should be made available to the reviewers

Line 109, which BacT/ALERT bottle was used?

Line 110-115, bacteriology methods poorly explained. Is there not a BARNARDS methods paper that could be cited? I find it a confusing mix of detail and high level summary i.e. was 0.2 ml blood really inoculated onto CBA before gram staining? It seems strange to be so precise about this volume and not detail whether oxidase and catalase were used to classify bacteria. This is why it would be useful to be able to cite a detailed BARNARDS method, or upload the standardised SOP as a supplemental material for reproducibility

Line 122 is entitled bacteriology, however this begins at 110

Line 123: CU – first use of acronym?

Is there anything about these methods that might have biased against the isolation of group B strep? I agree that various African settings have found lower levels than predicted of GBS, but GBS are represented in the Okomo review and the Malawi series

I think the bioinformatic methods are very well written and clearly well executed

Isolate data profiles and statistical analysis – I cannot see that any statistical analysis was undertaken outwith the maximum likelihood models used to reconstruct the phylogeny

Line 184 "scores" implies a summary statistic, whereas this was just the count of each was it not? What was done with the isolate metadata? Although clear from the results figures, this should be made clear

Results

Throughout the results there is slightly loose use of epidemiological terms such as prevalence. My impression is the authors are relating numbers of bacteria from a site to the numbers of that species within the collection rather than denominating by the number of cases of neonatal sepsis in a specific geographic location, which would be more informative

Line 203 the definition of a contaminant should be clarified in the methods, not in the results

Line 210 more detail required on "variable loss of viable isolates" necessary and list as a limitation – does this have implications for more fastidious organisms?

Line 227 a bit odd to talk about the 10 most common then only list 5

Line 230-231 – these are some of the most interesting data, surely describing the highlights of the site to site differences in Fig 2 should be a priority?

Klebsiella section – please go into more detail about frequency and geographical distribution of O-types as this has implications for vaccine design, whereas K type is unlikely to

Line 316 – what is a high virulence profile?

Line 373 – "The highest prevalence..." simply state that the greatest number of E.coli derived from Nigeria followed by Rwanda. Prevalence not helpful

Line 376 – at 371 it states there were "at least 90 STs" and at 376 37, which is it?

Line 415 – I know little about Serratia biology. Is it known to be clonal (i.e. is this expected) or is this a surprise? Define closely related.

Line 423 – They were most prevalent – in what sense? The greatest proportion of your collection at 49% was from Pakistan, but how did the frequency of isolation of Enterobacter relate to the populations being sampled?

Line 458-62 – Salmonella serovars should not be italicised (line 459-60) but should start with a capital (Line 462) and at line 461-2 the authors move into present tense

Figures – all the legends are weak. These are very complex figures with a vast amount of data in them. To make them useful to people who are not bioinformaticians (i.e. the majority of neonatologists, who may wish to read this paper), the reader should be walked through the highlights, from title to methods to results

Fig 1 - BB – explain acronym in legend

Fig 2 – suggest putting a-c, then d-f in rows, then gh and ij, then you can expand the legend and increase the font size. In legend, do you mean data not shown rather than now shown?

Fig 3-5 are too small to evaluate as presented in this pdf, especially the "c" figures for each. Each needs 3 whole pages, there is so much to each one and this relates to one of my major criticisms. It may be that the authors have higher resolution files they could upload, but I find as I zoom in they become blurred and difficult to appreciate

Discussion/Conclusions

Despite Barnards having a considerable focus on AMR, the AMR story is not discussed in great depth in this section, or placed in a global context, this should be done

Line 473 agree, but this manuscript does not describe incidence and RFs either

Line 474 neonatal sepsis belatedly adopted here

Line 478-480 – I agree which makes it a shame that this manuscript has only described genomes

Line 482- 524 this needs extensive revision. Quite a bit of repetition of the results

Variable loss of isolates should be discussed in limitations

Line 526-534 about staph good, but it is a bit of a non-sequitur to then report "most notably Group B Strep was not found..." as GBS is of course a strep not a staph

Line 539-542 – has this been lifted from another paper? The authors have not reported mortality in this paper, so this is not a limitation

Line 544 – 550 – this conclusion is odd – there is almost nothing in the first two sentences that could not have been gleaned from classical microbiology – what value has the genomics added? I agree that the resolution provided by WGS facilitates deeper epidemiological analysis, but that has not really been undertaken here

Author Rebuttal to Initial comments

Reviewer #1 (Remarks to the Author):

The paper "Genomic analysis of bacteria causing infant sepsis in low- and middle-income countries" is a description of the organisms isolates from the Gates funded BARNARDS (Burden of Antibiotic Resistance in Neonates in Developing Societies) study and is a comprehensive description of the organisms associated with sepsis in several sites. The study is excessively broad and lacks any real focus and whilst I appreciate the effort and the detailed analysis the paper represents a huge data dump with no real story or focus. Such multi-pathogen papers are hard to publish as these fields tend to be dominated by groups working on specific organisms and whilst the study is an excellent framework and whilst the analysis is good is nothing ground-breaking and the genomic analysis does not seem to add anything particularly new. There are no risk factors presented (as suggested) and no link with clinical data (as also suggested) or indeed any other variable that was likely collected and in its current state the paper lacks a message and adds little to these fields. There are some interesting insights into these organisms and a more detailed/specific focus on one of them would likely be a more interesting and novel addition to the field. In summary the paper, whilst a huge effort, does not really add anything new to our knowledge of these organisms (apart from scale). Without some additional clinical or epidemiological data, it is a collection of trees and some largely illegible diagrams of AMR genes. It is a real pity as a some form of focus and it could rival the GEMS study or other large Gates studies conducted in LMICs.

Sands et al RESPONSES to reviewer#1 in italics:

“excessively broad and lacks any real focus”: *We acknowledge this concern, and the major change in the revised manuscript has been to incorporate our microbiology phenotype dataset to focus on both antimicrobial resistance and the degree of genomic diversity causing neonatal sepsis.*

“There are no risk factors presented (as suggested) and no link with clinical data (as also suggested)”: *We agree that the previous manuscript submission, was lacking clinical data and therefore the potential impact of this work was limited. To align with our genomics/phylogeny & now AMR profiling, we have integrated two distinct clinical datasets. The onset of sepsis, and the outcome of sepsis. We integrate exploratory statistical analyses into the results sections. We also now clearly state that an epidemiological study into the risk factors for biological sepsis and mortality following sepsis are to be described elsewhere. An early draft of this was made available in the submission portal.*

“There are some interesting insights into these organisms and a more detailed/specific focus on one of them would likely be a more interesting and novel addition to the field” *We have completely restructured the results section to acknowledge the concern of a ‘data dump’ and to enhance details/specific focus on those bacterial species that were particularly isolated during the study. We do however feel that encapsulating and describing the genome collection as a whole is extremely important. The scale of this study is also one of the novel elements and we emphasise there*

is a large diversity of bacteria (74 species) causing neonatal sepsis, with clear geographic/local clusters occurring in the countries studied.

“Without some additional clinical or epidemiological data, it is a collection of trees and some largely illegible diagrams of AMR genes.” We agree with the reviewer and the scale of the dataset was at an expense of the clinical picture. We have restructured and rewritten the majority of the manuscript to incorporate clinical data. The genomics analysis and accompanying phylogenetic trees have been stripped back to only show those of importance to the main body of the text. We opted to include a large summary table linking epidemiological, clinical, phenotypic and genotypic data for the key species characterised during the study – to again clearly emphasise the large diversity. As such, the first 2 main figures and the inclusion of table 1 into the main manuscript are additions in this submission. Antimicrobial resistance was one of the fundamentals behind the BARNARDS project, and this is now emphasised by reporting on the microbiological phenotype – which by itself, was a huge microbiological component within this study. The description of AMR genes has been incorporated into the manuscript across multiple sections, notably into a section of “Antimicrobial resistance: phenotype and genotype of bacteria causing neonatal sepsis” in lines 130-172, but we have also removed any illegible diagrams including heatmaps, instead offering a supplementary table for readers.

Reviewer #2 (Remarks to the Author):

Sands et al RESPONSES to reviewer#2 in italics:

In “Genomic analysis of bacteria causing infant sepsis in LMIC” Sands and colleagues describe an analysis of 1067 bacterial genomes from the Barnards consortium. The collection represents a major, recent global snapshot and the genomic analysis is well performed and well described. In that sense the work in this manuscript should be commended and recognised as important.

The manuscript is not without significant flaws. The description is so “high level” as to render it uninformative. This is evidenced by the fact that there is no concluding line to the abstract, which instead peters out after the results summary. I think there are several reasons for this. Firstly, the authors have simply situated several key genera within their collection in a global context, rather than placing them more fully in the clinical context in which they were isolated, that of neonatal sepsis in diverse LMICs. The authors rightly highlight that a strength of BARNARDS is that it goes from clinical observation through classical microbiology to genomics, yet all that is really on display here is the genomics, and there is so much diversity that none of the undoubtedly important detail emerges. For example were specific lineages/virulence factors/AMR patterns associated with elevated mortality? Or specific outbreaks?

We agree with reviewer#2 in a similar manner to reviewer#1 – the analysis and description of results in the previous submission was exhaustive and repetitive. We have completely restructured this manuscript, where we hope the reader be able to understand that one of our reasons for undertaking a large volume of WGS was to allow complete characterisation of all isolates that we were able to successfully retrieve at Cardiff University from the LMICs. We did not want to select or focus on certain species at the beginning of the project, although we completely agree that the previous iteration did not discuss the clinical context, and very little discussion was had on antimicrobial resistance – a clear driving force behind the project. The addition of the phenotype, MIC by agar dilution & interpretation via EUCAST guidelines has allowed this revision to not only focus on the large genomic

variability, but we have been able to categorically state the level of AMR across a large selection of antibiotics. We now therefore blend classical microbiology with genomics, and ultimately show high rates of AMR across multiple antibiotic classes, including first line treatments for neonatal sepsis, but also reveal the co-occurrence of several AMR lineages. As such, we were able to re-write the abstract sufficiently.

For Klebsiella pneumoniae, Escherichia coli and Staphylococcus aureus, we felt it important show genomic context. We did complete an extensive literature search for available data in in both LMIC and globally. We incorporated those accordingly into our results, and we do feel that providing a genomic context in addition to a focus on neonatal sepsis, provides further information beyond these STs causing infection in LMIC. We do however acknowledge that this study focus is within LMIC and we have extended these conversations and literature in the discussion (for example sections beginning on lines 309, 334, and 348).

To address the comment concerning specificity of results in the context of clinical data, we have included data on both the onset of sepsis and the outcome of sepsis. We performed statistical analyses to allow us to ascertain whether associations occurred between microbiological/genomic outputs and the onset/outcome of sepsis. These analyses are spread throughout the results section. For example, we explore the resistance phenotype in relation to the onset/outcome of sepsis. Specifically, we also look at genetic traits such as MRSA, SA-PVL virulence, Klebsiella virulence and ask the question whether isolates with these genomic traits associate with a quicker onset and or a fatal outcome. All statistical analyses are detailed in a supplementary table, with the data available in the manuscript for those discussed. We clearly state our limitations with the statistics employed within this study. There is evidence of multiple outbreaks during BARNARDS, as clearly indicated by the phylogeny alone. We do however feel that detailed exploration of these is outside of the scope of this manuscript. Furthermore, we have accompanying microbiological and genomics data for isolates from the clinical environment, as another aspect of BARNARDS involved regular swabbing of the wards and clinical environment. As such, defined outbreaks are currently being investigated for publication elsewhere and this now been clearly emphasised in the discussion. Of course, we do make reference in the manuscript, that for some species, we observed less genomic diversity, and these are likely to represent local clusters. We provide the phylogenetic trees for those species that we refer to and the reader can see the core genome analysis, perhaps as in introduction before detailed SNP analysis is to be revealed in addition to complementary long read sequencing.

It feels to me as if the authors have got lost in trying to summarise this vast data set and consequently not pulled out any interesting narratives – consequently the discussion largely restates the results and does not discuss the data. They highlight a couple of potentially interesting stories i.e. regarding K. michiganensis and Staph aureus and then do not go into any further detail. I wonder if a better approach would be to submit one manuscript per major genus for the most common genera and then go into much greater depth. Instead of submitting 3-4 panel figures that are difficult to interpret even when one zooms in (perhaps the fault of the manuscript submission process?), the authors could then provide 3 large, individual high resolution figures per genus. They could then drill into genus and geography specific stories, comparing and contrasting differences within and between study sites and make much more of the clinical metadata associated with each isolate. I would have thought a Klebsiella paper on its own would be worthy of consideration by a Nature Journal.

We agree with the reviewer in relation to the quality of the discussion and have since revised the discussion to avoid restating our results at the expense of contextualising our findings. In relation to specific details of outbreaks, as mentioned, due to the volume of local clusters, some larger than others, we feel this warrants further investigation outside of this manuscript with a particular hospital/site focused approach, as we have accompanying microbiology, which ultimately varies according to the clinical site. The major revisions to the results are as follows:

1. *Inclusion of the phenotype and a focus on both AMR profiles and ARG traits.*
2. *A focus on the results of common genera as recommended; K. pneumoniae, S. aureus & E. coli, instead of listing all genera and limiting reporting to a few lines each. We also encapsulate the large diversity of species causing sepsis from the beginning of the results & the availability of genomes across as many species as possible is something we feel strongly. Although the reviewer makes a good point in relation to splitting this manuscript across major genera, we would like to this manuscript to remain as one comprehensive analysis. In order to set the scene more efficiently, Table 1 now provides an overview of the diversity, key phenotypic and genomic traits whilst also allowing us to minimise over reporting phylogenetic trees in the supplementary data. Furthermore, these are limited to those we specifically make reference to in the text – for example suggesting potential clonality for future investigation.*
3. *Following the overview of diversity and reporting on the AMR within the study across species, and in accordance with our findings & neonatal/global context analyses, we have focused on reporting the occurrence of K. pneumoniae, S. aureus & E. coli sepsis during the LMICs studied.*
4. *We have included clinical metadata into the main body of the manuscript, and we feel analysis of the microbiology in relation to onset/outcome of sepsis has greatly improved the story of the article by providing some clinical background and context in addition to the isolate characterisation.*

Lastly BARNARDS is supposed to have an AMR lens and indeed AMR genes are dutifully reported, however AMR is barely commented upon in the discussion. Again this theme suffers from being so high level for each genus.

We completely agree and hope this concern has been addressed by the inclusion of the MIC phenotype dataset. We have therefore increased our focus towards AMR and the associated clinical problem within this manuscript.

There are various ways in which I think the manuscript needs extensive editing (see below), but I think its greatest weakness is that the authors try to compress too much data into one overarching manuscript and consequently fail to say anything especially interesting.

Introduction:

The manuscript reads as if written by a bioinformatician – the bioinformatic methods and results are well described, but there is a disconnect between this is and the epidemiology and broader clinical context. For example, the introduction is poorly written i.e. line 47 “Infectious diseases mostly affect children in LMC...” is weak – there is a greater burden of severe bacterial infections in LMIC and clinical outcomes are worse, but there is no shortage of infections in high income settings, indeed 50% of primary care consultations in the UK are still related to suspected infection.

The introduction has been largely rewritten to align with the new focus of the article. We do acknowledge the concern the reviewer had and we have been cautious in our statements.

Why have they chosen the term “infant sepsis” rather than “neonatal sepsis”? There may be a good reason and if so this should be specified. Certainly “neonatal sepsis” has more traction and would gain the interest of neonatologists. Looking at supplemental fig 1 it looks as if participants recruited up to 60 days so it is not a strict neonatal population?

Following further discussion, we have decided upon the term neonatal sepsis for all of our manuscripts. We agree with the reviewer here and feel like the inclusion of the word 'neonatal' will gain more traction.

Line 56-58 poorly written and should be re-phrased, not sure what the authors are trying to say. As I cannot understand the point made in this sentence, the next one is meaningless to me.

Has been removed and rewritten.

Line 60-62, the authors should reflect on and cite the recent Lancet ID systematic review on neonatal sepsis in SSA (Okomo et al 2019, PMID: 31522858), also the single centre data from Iroh-Tam et al CID 2018 (PMCID: PMC6579959)

Discussions of the Okomo et al review have been included into both the introduction and discussion. We did not specifically include the citation for the Iroh-Tam et al study in Malawi over other single centre studies already cited, however we can build this into the discussion at the reviewer's request. The study by Iroh-Tam et al would be cited within lines 317-326. Furthermore, Iroh-Tam's reporting on increasing resistance to first line antibiotics could also suit citation in our adjacent study detailing antibiotic prescription/availability in the LMIC, current WHO guidelines, drug target attainment, and patient outcome. We thank the reviewer for addressing this.

Line 68, I dislike "uniquely" and generally claims of primacy should be discouraged

Agreed and removed

Methods

I would like to see a supplemental table with all the ethics committees listed and the relevant reference numbers or a reference to where these can be seen. Did Cardiff not provide ethics and thus sponsorship in addition to the local RECs?

Supplementary table added, and further details have been made available on the BARNARDS website.

Line 89-90 a lab SOP would not have provided an SOP for "the clinical diagnosis of sepsis" but for the laboratory processing of blood cultures and the subsequent identification of bacteria.

We agree with the reviewer and this has been edited to reflect.

What were the criteria for "clinical diagnosis of sepsis" I appreciate this may appear elsewhere, but the criteria for recognising a case of neonatal sepsis and then obtaining blood for culture should be presented in slightly more detail at line 106.

We have uploaded all SOPs and guidelines to a location on the BARNARDS website, and the URLs are made available within the manuscript

Is the lancet ID paper (ref 16) in press? If so say so, if not I would be cautious about citing it and the manuscript should be made available to the reviewers

We have edited the way in which we make reference to our adjacent studies, but we have not cited this within the reference list, as this is currently unpublished. Available manuscripts were uploaded to the portal upon submission. The Carvalho et al manuscript is still under preparation, but the 'epidemiology' and 'antibiotic dosing' manuscripts are available.

Line 109, which BacT/ALERT bottle was used?

Complete SOP uploaded to website for clarity

Line 110-115, bacteriology methods poorly explained. Is there not a BARNARDS methods paper that could be cited? I find it a confusing mix of detail and high level summary i.e. was 0.2 ml blood really inoculated onto CBA before gram staining? It seems strange to be so precise about this volume and not detail whether oxidase and catalase were used to classify bacteria. This is why it would be useful to be able to cite a detailed BARNARDS method, or upload the standardised SOP as a supplemental material for reproducibility

Likewise to the above response, in the absence of a BARNARDS methods paper, we have uploaded all SOPs and guidelines to a location on the BARNARDS website, and the URLs are made available within the manuscript

Line 122 is entitled bacteriology, however this begins at 110

Line 123: CU – first use of acronym?

Is there anything about these methods that might have biased against the isolation of group B strep? I agree that various African settings have found lower levels than predicted of GBS, but GBS are represented in the Okomo review and the Malawi series

Our limitations on GPB isolation has been clearly emphasised in this revision. We have also discussed this in lines 371-375. We also acknowledge other studies that were not included in the previous submission that evidence a lack of GBS, however we do acknowledge our limitations with the possible recovery of GBS.

I think the bioinformatic methods are very well written and clearly well executed

Isolate data profiles and statistical analysis – I cannot see that any statistical analysis was undertaken outwith the maximum likelihood models used to reconstruct the phylogeny

Line 184 “scores” implies a summary statistic, whereas this was just the count of each was it not?

What was done with the isolate metadata? Although clear from the results figures, this should be made clear

We agree with the reviewer and we have added a component of statistical analyses to this study. The results of which are peppered throughout the manuscript, but the methods are detailed in lines 486-499. We have also revised the format of the manuscript, so that the methods follow the discussion.

Results

Throughout the results there is slightly loose use of epidemiological terms such as prevalence. My impression is the authors are relating numbers of bacteria from a site to the numbers of that species within the collection rather than denominating by the number of cases of neonatal sepsis in a specific geographic location, which would be more informative

We have addressed this, and we now report on frequencies, rather than attempting to state prevalence for a dataset on biological sepsis of which analysed only a portion of by WGS.

Line 203 the definition of a contaminant should be clarified in the methods, not in the results

Moved to methods in lines 437-439.

Line 210 more detail required on “variable loss of viable isolates” necessary and list as a limitation – does this have implications for more fastidious organisms?

Line 227 a bit odd to talk about the 10 most common then only list 5

Line 230-231 – these are some of the most interesting data, surely describing the highlights of the site to site differences in Fig 2 should be a priority?

Klebsiella section – please go into more detail about frequency and geographical distribution of O-types as this has implications for vaccine design, whereas K type is unlikely to

Line 316 – what is a high virulence profile?

Line 373 – “The highest prevalence...” simply state that the greatest number of E.coli derived from Nigeria followed by Rwanda. Prevalence not helpful

Line 376 – at 371 it states there were “at least 90 STs” and at 376 37, which is it?

Line 415 – I know little about Serratia biology. Is it known to be clonal (i.e. is this expected) or is this a surprise? Define closely related.

Line 423 – They were most prevalent – in what sense? The greatest proportion of your collection at 49% was from Pakistan, but how did the frequency of isolation of Enterobacter relate to the populations being sampled?

Line 458-62 – Salmonella serovars should not be italicised (line 459-60) but should start with a capital (Line 462) and at line 461-2 the authors move into present tense

“loss of viable isolates” first eluded to in lines 83-85 has been further discussed as a limitation in the discussion, with particular reference to loss of fastidious organisms and with respect to recovery of GBS in lines 373-375.

We agree, and the figure this relates to, now figure 1 has been completely redone to provide an overview of the clinical sites (map) with an embedded bar graph displaying the 10 most frequently isolated species per site. Line 88, page 4 for figure 1.

The time to sepsis in the previous manuscript has since been removed from this submission as we feel this is more suited to one of our adjacent studies whereby, they examine risk factors to biological sepsis and mortality. We have opted to stick to the onset of sepsis as our clinical metadata, and this has been analysed at a species level throughout the paper.

Since this manuscript discusses the genomic characterisation of a subset of sepsis isolates, we have refrained from the use of prevalence throughout when referring to numbers of bacteria found. We have provided a comprehensive supplementary table 1 where the readers can find the complete microbiological dataset per site.

For the Klebsiella O:K serotype, we felt that we did not have sufficient justification and scope to explore this fully, at this time, although we have made reference of the potential contribution that this dataset can have in relation to vaccine development for Klebsiella neonatal infection. We have therefore removed that dataset from the phylogeny, but we do provide O:K serotyping within a larger isolate supplementary table, as we feel that access to this large genomics dataset will inform further work both within WGS in LMIC and within AMR and exploring effective management and treatment.

Figures – all the legends are weak. These are very complex figures with a vast amount of data in them. To make them useful to people who are not bioinformaticians (i.e. the majority of neonatologists, who may wish to read this paper), the reader should be walked through the highlights, from title to methods to results

Fig 1 - BB – explain acronym in legend

We have revised the large majority of figures and accompanying legends to provide additional information as outlined by the reviewers.

Fig 2 – suggest putting a-c, then d-f in rows, then gh and ij, then you can expand the legend and increase the font size. In legend, do you mean data not shown rather than now shown?

This figure has been removed from this manuscript and replaced by bar graphs detailing the percentage of isolates resistant to antibiotics tested. These are stacked bar graphs, coloured by bacterial species/group category to avoid excess categories.

Fig 3-5 are too small to evaluate as presented in this pdf, especially the “c” figures for each. Each needs 3 whole pages, there is so much to each one and this relates to one of my major criticisms. It may be that the authors have higher resolution files they could upload, but I find as I zoom in they become blurred and difficult to appreciate.

A lot of revision has taken place for all phylogenetic trees and associated genomic information. Additionally, the files are of a greater resolution.

Discussion/Conclusions

Despite Barnards having a considerable focus on AMR, the AMR story is not discussed in great depth in this section, or placed in a global context, this should be done

We agree with the reviewer and hope the revised manuscript addresses this. The section of phenotype-genotype results overview sets the scene for the burden of AMR in neonatal sepsis as found during this study

Line 473 agree, but this manuscript does not describe incidence and RFs either

This has been removed and edited.

Line 474 neonatal sepsis belatedly adopted here
We now use neonatal sepsis throughout

Line 478-480 – I agree which makes it a shame that this manuscript has only described genomes
Added phenotype allows integrated discussion, and results reporting when linking phenotype-genotype.

Line 482- 524 this needs extensive revision. Quite a bit of repetition of the results

We agree with the reviewer and the discussion has been majorly revised to avoid repetition of results.

Variable loss of isolates should be discussed in limitations

Line 526-534 about staph good, but it is a bit of a non-sequitur to then report “most notably Group B Strep was not found...” as GBS is of course a strep not a staph

Both sections in the discussion have been revised

Line 539-542 – has this been lifted from another paper? The authors have not reported mortality in this paper, so this is not a limitation

We have now clearly included mortality into the main text and analysis of the manuscript.

Line 544 – 550 – this conclusion is odd – there is almost nothing in the first two sentences that could not have been gleaned from classical microbiology – what value has the genomics added? I agree that the resolution provided by WGS facilitates deeper epidemiological analysis, but that has not really been undertaken here

The conclusion has been refocused to detail the main causes of neonatal sepsis, to state the levels of AMR, and to emphasise the wide range of ARG/virulent traits in diverse species. Notably, we end by stating that this work can inform further work investigating treatment options, and potentially vaccine development for neonatal sepsis in LMICs.

Decision Letter, first revision:

Dear Kirsty,

Thank you for your patience while your manuscript "Genomic epidemiology and patterns of antimicrobial resistance of neonatal sepsis in low- and middle-income countries" was under peer-review at Nature Microbiology. It has now been seen by 2 referees, whose expertise and comments you will find at the of this email. You will see from their comments below that while they find your

work of interest, some important points are raised. We are very interested in the possibility of publishing your study in Nature Microbiology, but would like to consider your response to these concerns in the form of a revised manuscript before we make a final decision on publication.

In particular, you will see that both referees are still concerned over the focus of the study. Specifically, referee #1 feels that "...there is a lack of detail for each organism which limits the potential scope of the paper." The referee also mentions that "...a solid epidemiological description of the clinical/outcome data in its entirety should be the key aim of investigation. Then, a focused analysis of each organism would be of larger value to the fields...". Referee 2 thinks the way the diversity at each site is presented could be improved, and says "it is surprising that the authors do not reflect on the observed patterns of resistance more in the discussion in terms of their implications for either empirical or culture guided management of neonatal sepsis." In the comments to the editor, this referee also mentions that "An enormous amount of data is presented as a surveillance piece rather than as a discreet piece of hypothesis driven research, so the challenge is to conjure a narrative to bring it together cohesively. If they can develop the themes of intra species diversity and AMR still further I think it will achieve this." Editorially, we would ask you to reflect on the write-up, and to try to improve the narrative of the manuscript according to the referees' suggestions, specifically developing the themes of intraspecies diversity and AMR further. The rest referees' reports are clear and the remaining issues should be straightforward to address.

If you have not done so already please begin to revise your manuscript so that it conforms to our Article format instructions at <http://www.nature.com/nmicrobiol/info/final-submission/>

The usual length limit for a Nature Microbiology Article is six display items (figures or tables) and 3,000 words. We have some flexibility, and can allow a revised manuscript at 3,500 words, but please consider this a firm upper limit. There is a trade-off of ~250 words per display item, so if you need more space, you could move a Figure or Table to Supplementary Information.

Some reduction could be achieved by focusing any introductory material and moving it to the start of your opening 'bold' paragraph, whose function is to outline the background to your work, describe in a sentence your new observations, and explain your main conclusions. The discussion should also be limited. Methods should be described in a separate section following the discussion, we do not place a word limit on Methods.

Nature Microbiology titles should give a sense of the main new findings of a manuscript, and should not contain punctuation. Please keep in mind that we strongly discourage active verbs in titles, and that they should ideally fit within 90 characters each (including spaces).

We strongly support public availability of data. Please place the data used in your paper into a public data repository, if one exists, or alternatively, present the data as Source Data or Supplementary Information. If data can only be shared on request, please explain why in your Data Availability Statement, and also in the correspondence with your editor. For some data types, deposition in a public repository is mandatory - more information on our data deposition policies and available repositories can be found at <https://www.nature.com/nature-research/editorial-policies/reporting->

standards#availability-of-data.

Please include a data availability statement as a separate section after Methods but before references, under the heading "Data Availability". This section should inform readers about the availability of the data used to support the conclusions of your study. This information includes accession codes to public repositories (data banks for protein, DNA or RNA sequences, microarray, proteomics data etc...), references to source data published alongside the paper, unique identifiers such as URLs to data repository entries, or data set DOIs, and any other statement about data availability. At a minimum, you should include the following statement: "The data that support the findings of this study are available from the corresponding author upon request", mentioning any restrictions on availability. If DOIs are provided, we also strongly encourage including these in the Reference list (authors, title, publisher (repository name), identifier, year). For more guidance on how to write this section please see:

<http://www.nature.com/authors/policies/data/data-availability-statements-data-citations.pdf>

To improve the accessibility of your paper to readers from other research areas, please pay particular attention to the wording of the paper's opening bold paragraph, which serves both as an introduction and as a brief, non-technical summary in about 150 words. If, however, you require one or two extra sentences to explain your work clearly, please include them even if the paragraph is over-length as a result. The opening paragraph should not contain references. Because scientists from other sub-disciplines will be interested in your results and their implications, it is important to explain essential but specialised terms concisely. We suggest you show your summary paragraph to colleagues in other fields to uncover any problematic concepts.

If your paper is accepted for publication, we will edit your display items electronically so they conform to our house style and will reproduce clearly in print. If necessary, we will re-size figures to fit single or double column width. If your figures contain several parts, the parts should form a neat rectangle when assembled. Choosing the right electronic format at this stage will speed up the processing of your paper and give the best possible results in print. We would like the figures to be supplied as vector files - EPS, PDF, AI or postscript (PS) file formats (not raster or bitmap files), preferably generated with vector-graphics software (Adobe Illustrator for example). Please try to ensure that all figures are non-flattened and fully editable. All images should be at least 300 dpi resolution (when figures are scaled to approximately the size that they are to be printed at) and in RGB colour format. Please do not submit Jpeg or flattened TIFF files. Please see also 'Guidelines for Electronic Submission of Figures' at the end of this letter for further detail.

Figure legends must provide a brief description of the figure and the symbols used, within 350 words, including definitions of any error bars employed in the figures.

Please include a statement before the acknowledgements naming the author to whom correspondence and requests for materials should be addressed.

Finally, we require authors to include a statement of their individual contributions to the paper -- such as experimental work, project planning, data analysis, etc. -- immediately after the acknowledgements. The statement should be short, and refer to authors by their initials. For details please see the Authorship section of our joint Editorial policies at http://www.nature.com/authors/editorial_policies/authorship.html

* include a point-by-point response to any editorial suggestions and to our referees. Please include your response to the editorial suggestions in your cover letter, and please upload your response to the referees as a separate document.

* ensure it complies with our format requirements for Letters as set out in our guide to authors at www.nature.com/nmicrobiol/info/gta/

* state in a cover note the length of the text, methods and legends; the number of references; number and estimated final size of figures and tables

* resubmit electronically if possible using the link below to access your home page:

{REDACTED}

*This url links to your confidential homepage and associated information about manuscripts you may have submitted or be reviewing for us. If you wish to forward this e-mail to co-authors, please delete this link to your homepage first.

Please ensure that all correspondence is marked with your Nature Microbiology reference number in the subject line.

Nature Microbiology is committed to improving transparency in authorship. As part of our efforts in this direction, we are now requesting that all authors identified as 'corresponding author' on published papers create and link their Open Researcher and Contributor Identifier (ORCID) with their account on the Manuscript Tracking System (MTS), prior to acceptance. This applies to primary research papers only. ORCID helps the scientific community achieve unambiguous attribution of all scholarly contributions. You can create and link your ORCID from the home page of the MTS by clicking on 'Modify my Springer Nature account'. For more information please visit www.springernature.com/orcid.

We hope to receive your revised paper within three weeks. If you cannot send it within this time, please let us know.

Reviewer Expertise:

Referee #1: {REDACTED}

Referee #2: {REDACTED}

Reviewers Comments:

Reviewer #1 (Remarks to the Author):

This is the second time I have reviewed this paper. This is a remarkable amount of work and it is a well-structured and highly valuable study. But my opinion hasn't changed; the paper should be commended but it provides no new insights into neonatal sepsis. The genomics is interesting but there is a lack of detail for each organism which limits the potential scope of the paper. As I said before, a solid epidemiological description of the clinical/outcome data in its entirety should be the key aim of investigation. Then, a focused analysis of each organism would be of larger value to the fields than this amalgamation of all sequence data. As it stands it is difficult to join all the pieces together and comparing the different organisms in one go makes little sense.

Reviewer #2 (Remarks to the Author):

I would like to thank the authors for engaging so thoroughly with my suggestions for strengthening the manuscript. This amended version is greatly improved. I have two general points and a few specific minor requests for further clarifications.

1. Whilst intraspecies diversity across the study is well described and well presented in the trees, the diversity at each site does not come across terribly strongly either in the text or the trees
2. It is surprising that the authors do not reflect on the observed patterns of resistance more in the discussion in terms of their implications for either empirical or culture guided management of neonatal sepsis. Clearly they will to do this in great detail in the lancet ID manuscript, but there is surprisingly little here and a high level reflection in one paragraph of the discussion would be valuable to make this paper more complete. Are patterns the same in African and Asian sites? If not are they broadly similar across the continents ? I would like to see the paragraph at line 328 of the discussion expanded without compromising the lancet paper

Line 39-42: This is quite a bland line, because the authors are trying to compress so much information in, could it be expanded a bit beyond saying "some bugs were diverse and some where not" ?

Line 86-87: this is better explained in the methods, could you expand it a little here please?

Line 88: are, not is

Line 131-2: you start off talking about GNB resistance and flip to drug activity, could you simply say whereas they remained susceptible to...

Line 139-140: clumsily phrases, please expand a bit to make easier to understand

Line 140-141: "...likely to cause..." this is a statistical association, you have not demonstrated causality

Line 190: what does the (387) refer to?

Line 348-354: not sure what point this paragraph is trying to make, in particular line 349-351 badly phrased

Line 359-360: rephrase "...not always matched. ...did not always match??"

Author Rebuttal, first revision:

Reviewer #1 (Remarks to the Author):

This is the second time I have reviewed this paper. This is a remarkable amount of work and it is a well-structured and highly valuable study. But my opinion hasn't changed; the paper should be commended but it provides no new insights into neonatal sepsis. The genomics is interesting but there is a lack of detail for each organism which limits the potential scope of the paper. As I said before, a solid epidemiological description of the clinical/outcome data in its entirety should be the key aim of investigation. Then, a focused analysis of each organism would be of larger value to the fields than this amalgamation of all sequence data. As it stands it is difficult to join all the pieces together and comparing the different organisms in one go makes little sense.

Response to Reviewer #1

We would like to thank Reviewer #1 for the valuable suggestions, and we have taken these on board. we have included metadata to contextualise isolates and related neonates' outcomes/onset of sepsis. We also have other manuscripts in preparations that solely focus on clinical and epidemiological data on neonatal sepsis and mortality following sepsis; an international prospective birth cohort study (and an updated manuscript has been made available to reviewers upon resubmission). To help guide the reader through the various analyses, at the beginning of each results section we have added a few lines to describe the background and aim of the analysis performed. Additionally, lines 89-93 have been included to help set the scene for the reader, particularly with regards to the fundamental aims of BARNARDS, and the key aim for this manuscript was to ultimately collect and characterize all Gram-negative bacteria causing neonatal sepsis. Within BARNARDS, it was always the intension to characterise all Gram-negative bacteria causing sepsis and perform whole genome sequencing and susceptibility profiling to reveal the extent of resistance/resistant genetic traits across all Gram-negative species for all clinical sites in Africa and South-Asia and not pre-select.

We have decided to remove the Gram-positive dataset from this manuscript. This has provided additional space to improve the focus and scope to the paper. To achieve this focus, we have tailored the manuscript around two main themes:

- 1. Interspecies and intraspecies diversity*
- 2. Antimicrobial resistance*

We agree with reviewer #1 in that the results would benefit from greater inclusion of clinical data and this has been incorporated further into the beginning of the results section to allow contextualisation of the pathogens causing sepsis (we regard to onset and outcome). Figure 1 has been edited to include a comprehensive flow diagram (Figure 1a) detailing the recruitment at each site, the numbers of clinical sepsis, culture confirmed sepsis and the n= of missing data. All numbers are available overall and per clinical site. Furthermore, Figure 1b now shows a nested pie chart per clinical site summarising the onset of sepsis (confirmed) cases and a chart summarising the outcome data per continent. We then go provide a summary linking the pathogens causing sepsis to the clinical data; onset/outcome (lines 106-112).

Reviewer #2 (Remarks to the Author):

I would like to thank the authors for engaging so thoroughly with my suggestions for strengthening the manuscript. This amended version is greatly improved. I have two general points and a few specific minor requests for further clarifications.

1. Whilst intraspecies diversity across the study is well described and well presented in the trees, the diversity at each site does not come across terribly strongly either in the text or the trees
2. It is surprising that the authors do not reflect on the observed patterns of resistance more in the discussion in terms of their implications for either empirical or culture guided management of neonatal sepsis. Clearly they will to do this in great detail in the lancet ID manuscript, but there is surprisingly little here and a high level reflection in one paragraph of the discussion would be valuable to make this paper more complete. Are patterns the same in African and Asian sites? If not are they broadly similar across the continents ? I would like to see the paragraph at line 328 of the discussion expanded without compromising the lancet paper

Response to Reviewer #2

*We would like to thank Reviewer #2 for the valuable suggestions, and we have taken these all on board. Firstly, we have made considerable edits to the text in light of the intraspecies diversity section, particularly when analysing data between the clinical sites. We now have a section (lines 114-142) introducing interspecies and intraspecies diversity of the total Gram-negative bacteria found during the study, which includes a comparison of other Gram-negative bacteria and the sequence types found at the different clinical sites, for *Enterobacter* and *Acinetobacter* for example. Furthermore, the *K. pneumoniae* and *E. coli* sections have been edited and now include more granular details comparing the intraspecies diversity between the clinical sites (lines 223-283 and 285-315 respectively).*

We have increased our focus to reporting on antimicrobial resistance in several linking areas within this manuscript. We agree with reviewer #2 and we have increased analysis towards reporting antimicrobial resistance rates. In the “Antimicrobial resistance of pathogens causing neonatal sepsis” section of

results, we now make reference to the n=% of Gram-negative bacteria that are resistant to both ampicillin and gentamicin as the recommended first line treatment. We also include a paragraph in the discussion (323-328) summarising this information. We reveal the rate of Gram-negative bacteria carrying carbapenemase genes is higher in South-Asia throughout the results. Finally, our concluding paragraph also makes reference that 54% of isolated bacteria were resistant to at least one antibiotic for four or more antibiotic classes highlighting multidrug resistance in Gram-negative bacteria causing neonatal sepsis.

Line 39-42: This is quite a bland line, because the authors are trying to compress so much information in, could it be expanded a bit beyond saying “some bugs were diverse and some where not” ?

We agree with the reviewer and have edited the abstract to avoid compressing too much information. The abstract now lists the dominant Gram-negative bacteria causing neonatal sepsis. We then move to a sentence revealing the extent of antimicrobial resistance

Line 86-87: this is better explained in the methods, could you expand it a little here please?

The Gram-positive dataset has been removed from the current revised manuscript to allow increased focus into Gram-negative bacteria and mechanisms of resistance. We do however include more information about the Gram-positives between lines 89-93.

Line 88: are, not is

This section has been edited

Line 131-2: you start off talking about GNB resistance and flip to drug activity, could you simply say whereas they remained susceptible to...

We agree with the reviewer and we have edited this to susceptible (lines 154-155).

Line 139-140: clumsily phrases, please expand a bit to make easier to understand

We agree with the reviewer and we have edited this section, removed the vague phrase and focused to associations between cephalosporin resistance and onset (lines 161-164).

Line 140-141: "...likely to cause..." this is a statistical association, you have not demonstrated causality

We agree and this has been edited (line 163)

Line 190: what does the (387) refer to?

These were indications of the word count and have all been removed.

Line 348-354: not sure what point this paragraph is trying to make, in particular line 349-351 badly phrased

We agree with the reviewer – this section has been removed from the discussion

Line 359-360: rephrase "...not always matched. ...did not always match??"

As the Gram-positive dataset has been removed, this has been deleted from the manuscript.

Decision Letter, second revision:

Dear Kirsty,

Thank you for your patience while your manuscript "Characterization of antimicrobial resistant Gram-negative bacteria that cause neonatal sepsis in seven low and middle-income countries" was under peer review at Nature Microbiology. I am delighted to say that we can in principle offer to publish it.

Nature Microbiology offers a transparent peer review option for new original research manuscripts submitted from 1st December 2019. We encourage increased transparency in peer review by publishing the reviewer comments, author rebuttal letters and editorial decision letters if the authors agree. Such peer review material is made available as a supplementary peer review file. **Please state in the cover letter 'I wish to participate in transparent peer review' if you want to opt in, or 'I do not wish to participate in transparent peer review' if you don't.** Failure to state your preference will result in delays in accepting your manuscript for publication.

Please note: we allow redactions to authors' rebuttal and reviewer comments in the interest of confidentiality. If you are concerned about the release of confidential data, please let us know specifically what information you would like to have removed. Please note that we cannot incorporate

redactions for any other reasons. Reviewer names will be published in the peer review files if the reviewer signed the comments to authors, or if reviewers explicitly agree to release their name. For more information, please refer to our [FAQ page](https://www.nature.com/documents/nr-transparent-peer-review.pdf).

In recognition of the time and expertise our reviewers provide to Nature Microbiology's editorial process, we would like to formally acknowledge their contribution to the external peer review of your manuscript. For those reviewers who give their assent, we will be publishing their names alongside the published article.

I appreciate this email is long and recommend that you print it and use it as a checklist, reading it carefully to the end, in order to avoid delays to publication down the line.

Please note that we will be considering your paper for publication as an ARTICLE in our pages.

Specific points:

In particular, while checking through the manuscript and associated files, we noticed the following specific points which we will need you to address:

1. Figures. For Figure 1, please have Figure 1A and Figure 2A as two separate figures, as they are quite large already on their own. Please make sure to renumber the other figures and the text accordingly. In total you would have 6 main figures.

2. Supplementary information. All Supplementary Information must be submitted in accordance with the instructions in the attached Inventory of Supporting Information, and should fit into one of three categories: EXTENDED DATA (ED); SUPPLEMENTARY INFORMATION (SI); and SOURCE DATA. Below are detailed instructions on how to format each category. For your paper, we suggest that you do the following:

a. Supplementary information (SI): your study will have the 'Supplementary online data' as SI. Please submit all SI as a separate pdf file. All supplementary materials need to be assembled into a single file, including all tables (excluding those that are excessively large). In the Supplementary Information file, figure legends should be immediately below each figure and the pages should be numbered.

b. Source data: this format should be used to display source data linked to the main figures and ED figures.

We strongly encourage you include as much additional raw data underlying the graphs in the main and ED figures as possible. These data should be supplied as Excel tables, one file per main or ED figure, and should be clearly labeled and presented in a way that individual experiments are identifiable (for example, across a time course if applicable).

3. Data Availability statement. The data availability statement should clearly refer to all of the source data provided in the manuscript (more instructions on how to write this section can be found in the general formatting guidelines below).

4. Reporting checklist. Please revise this document according to the instructions found in the annotated PDF attached to this message and send in a final version with your article. The final reporting checklist will be published with your manuscript. Specifically, please address the following:

- Please note that the (European Nucleotide Archive (ENA) accession no. (PRJEB33565) referenced in the manuscript is currently unavailable/ unreleased. A request has been made to the authors for the public release of this data, in the reporting summary.
- It was observed in the Human research participant's module of the reporting summary that the authors have stated the following "Individual clinical sites were responsible for obtaining ethical approval. All documentation can be shared on request. Cardiff University was the host/sponsor site and was responsible for overseeing the ethical procedures at the individual sites" in the ethical oversight section. Further, the manuscript references the availability of ethical approvals in Supplementary Table 8. A request has been made to the authors for the provision of this information in the corresponding section of the reporting summary as well.
- Please note that the authors have marked the clinical data section as "involved in the study" in the reporting summary; however, they have not provided the clinical trial registration number in the corresponding section or in the manuscript. Also, it was observed that the study does not involve clinical trials/clinical trial associated data. Hence, we have requested the authors to mark this module as "N/A", appropriately.

5. Data deposition. Please carefully check through the manuscript whether all different types of sequence data have been deposited in appropriate databases.

6. ORCID. We now require corresponding authors to provide an ORCID identifier, and would ask that you please provide one with your final submission (please also see below). There is a step during the upload of the information to our online system in which the number can be introduced.

General comments:

Wherever statistics have been derived (e.g., error bars, box plots, statistical significance), the legend needs to provide and define the n number (i.e., the sample size used to derive statistics) as a precise value (not a range), using the wording "n=X biologically independent samples/animals/independent experiments," etc. as applicable.

All error bars need to be defined in the legends (e.g., SD, SEM) together with a measure of centre (e.g. mean, median), and should be accompanied by their precise n number defined as noted above.

All violin plots need to be defined in the legends in terms of minima, maxima, centre, and percentiles, and should be accompanied by their precise n number defined as noted above.

The figure legends must indicate the statistical test used and if applicable, whether the test was one- or two-sided. A description of any assumptions or corrections such as tests of normality and adjustment for multiple comparisons must also be included.

For null hypothesis testing, please indicate the test statistic (e.g., F, t, r) with confidence intervals, effect sizes, degrees of freedom and P values noted.

Test results (e.g., p-values, q-values) should be given as exact values whenever possible and

appropriate, and confidence intervals noted.

Please indicate how estimates of effect sizes were calculated (e.g., Cohen's d , Pearson's r).

Please state in the legends how many times each experiment was repeated independently with similar results. This is needed for all experiments but is particularly important wherever representative experiments are shown. If space in the legends is limiting, this information can be included in a section titled "Statistics and Reproducibility".

For all bar graphs, the corresponding dot plot must be overlaid.

General points:

Please read carefully through all of the following general formatting points when preparing the final version of your manuscript, as submitting the manuscript files in the required format will greatly speed the process to final acceptance of your work.

Titles should give an idea of the main finding of the paper and ideally not exceed 90 characters (including spaces). We discourage the use of active verbs and do not allow punctuation.

The paper's summary paragraph (about 150-200 words; no references) should serve both as a general introduction to the topic, and as a brief, non-technical summary of your main results and their implications. It should start by outlining the background to your work (why the topic is important) and the main question you have addressed (the specific problem that initiated your research), before going on to describe your new observations, main conclusions and their general implications. Because we hope that scientists across the wider microbiology community will be interested in your work, the first paragraph should be as accessible as possible, explaining essential but specialised terms concisely. We suggest you show your summary paragraph to colleagues in other fields to uncover any problematic concepts.

Please include a data availability statement as a separate section after Methods but before references, under the heading "Data Availability". This section should inform readers about the availability of the data used to support the conclusions of your study. This information includes accession codes to public repositories (data banks for protein, DNA or RNA sequences, microarray, proteomics data etc...), references to source data published alongside the paper, unique identifiers such as URLs to data repository entries, or data set DOIs, and any other statement about data availability. At a minimum, you should include the following statement: "The data that support the findings of this study are available from the corresponding author upon request", mentioning any restrictions on availability. If

DOIs are provided, we also strongly encourage including these in the Reference list (authors, title, publisher (repository name), identifier, year). For more guidance on how to write this section please see:

<http://www.nature.com/authors/policies/data/data-availability-statements-data-citations.pdf>

Please supply the figures as vector files - EPS, PDF, AI or postscript (PS) file formats (not raster or bitmap files), preferably generated with vector-graphics software (Adobe Illustrator for example). Try to ensure that all figures are non-flattened and fully editable. All images should be at least 300 dpi resolution (when figures are scaled to approximately the size that they are to be printed at) and in RGB colour format. Please do not submit Jpeg or flattened TIFF files. Please see also 'Guidelines for Electronic Submission of Figures' at the end of this letter for further detail.

Please view http://www.nature.com/authors/editorial_policies/image.html for more detailed guidelines.

We will edit your figures/tables electronically so they conform to Nature Microbiology style. If necessary, we will re-size figures to fit single or double column width. If your figures contain several parts, the parts should be labelled lower case a, b, and so on, and form a neat rectangle when assembled.

Please check the PDF of the whole paper and figures (on our manuscript tracking system) VERY CAREFULLY when you submit the revised manuscript. This will be used as the 'reference copy' to make sure no details (such as Greek letters or symbols) have gone missing during file-transfer/conversion and re-drawing.

All Supplementary Information must be submitted in accordance with the instructions in the attached Inventory of Supporting Information, and should fit into one of three categories:

1. **EXTENDED DATA:** Extended Data are an integral part of the paper and only data that directly contribute to the main message should be presented. These figures will be integrated into the full-text HTML version of your paper and will be appended to the online PDF. There is a limit of 10 Extended Data figures, and each must be referred to in the main text. Each Extended Data figure should be of the same quality as the main figures, and should be supplied at a size that will allow both the figure and legend to be presented on a single legal-sized page. Each figure should be submitted as an individual .jpg, .tif or .eps file with a maximum size of 10 MB each. All Extended Data figure legends must be provided in the attached Inventory of Accessory Information, not in the figure files themselves.

2. **SUPPLEMENTARY INFORMATION:** Supplementary Information is material that is essential background to the study but which is not practical to include in the printed version of the paper (for example, video files, large data sets and calculations). Each item must be referred to in the main manuscript and detailed in the attached Inventory of Accessory Information. Tables containing large data sets should be in Excel format, with the table number and title included within the body of the table. All textual information and any additional Supplementary Figures (which should be presented with the legends directly below each figure) should be provided as a single, combined PDF. Please note that we cannot accept resupplies of Supplementary Information after the paper has been formally accepted unless there has been a critical scientific error.

All Extended Data must be called out in your manuscript and cited as Extended Data 1, Extended Data

2, etc. Additional Supplementary Figures (if permitted) and other items are not required to be called out in your manuscript text, but should be numerically numbered, starting at one, as Supplementary Figure 1, not SI1, etc.

3. SOURCE DATA: We strongly encourage you to provide source data for your figures whenever possible. Full-length, unprocessed gels and blots must be provided as source data for any relevant figures, and should be provided as individual PDF files for each figure containing all supporting blots and/or gels with the linked figure noted directly in the file. Numerical source data that underlie graphs are required for in vivo experiments and strongly encouraged generally. They should be provided in Excel format, one file for each relevant figure, with the linked figure noted directly in the file. They should be clearly labelled such that individual experiments and/or animals are labelled (for example, across a time course if applicable). For imaging source data, we encourage deposition to a relevant repository, such as figshare (<https://figshare.com/>) or the Image Data Resource (<https://idr.openmicroscopy.org>).

Please ensure that you retain unprocessed data and metadata files after publication, ideally archiving data in perpetuity, as these may be requested during the peer review and production process or after publication if any issues arise.

Please include any references for the Methods at the end of the reference list. Any citations in the Supplemental Information will need inclusion in a separate SI reference list.

It is a condition of publication that you include a statement before the acknowledgements naming the author to whom correspondence and requests for materials should be addressed.

Finally, we require authors to include a statement of their individual contributions to the paper -- such as experimental work, project planning, data analysis, etc. -- immediately after the acknowledgements. The statement should be short, and refer to authors by their initials. For details please see the Authorship section of our joint Editorial policies at http://www.nature.com/authors/editorial_policies/authorship.html

We will not send your revised paper for further review if, in the editors' judgement, the referees' comments on the present version have been addressed. If the revised paper is in Nature Microbiology format, in accessible style and of appropriate length, we shall accept it for publication immediately.

Please resubmit electronically

* the final version of the text (not including the figures) in either Word or Latex.

* publication-quality figures. For more details, please refer to our Figure Guidelines, which is available here: <https://www.nature.com/documents/NRJs-guide-to-preparing-final-artwork.pdf>

* Extended Data & Supplementary Information, as instructed

* a point-by-point response to any issues raised by our referees and to any editorial suggestions.

* any suggestions for cover illustrations, which should be provided at high resolution as electronic

files. Please note that such pictures should be selected more for their aesthetic appeal than for their scientific content. I am sure you will understand that we cannot make any promise as to whether any of your suggestions might be selected for the cover of Nature Microbiology.

Please use the following link to access your home page:

{REDACTED}

* This url links to your confidential homepage and associated information about manuscripts you may have submitted or be reviewing for us. If you wish to forward this e-mail to co-authors, please delete this link to your homepage first.

Please also send the following forms as a PDF by email to microbiology@nature.com.

* Please sign and return the [Licence to Publish form](http://www.nature.com/documents/snl-ltp.docx) .

* Or, if the corresponding author is either a Crown government employee (including Great Britain and Northern Ireland, Canada and Australia), or a US Government employee, please sign and return the [Licence to Publish form for Crown government employees](http://www.nature.com/documents/snl-ltp-crown.docx), or a [Licence to Publish form for US government employees](http://www.nature.com/documents/snl-ltp-govus.docx).

* Should your Article contain any items (figures, tables, images, videos or text boxes) that are the same as (or are adaptations of) items that have previously been published elsewhere and/or are owned by a third party, please note that it is your responsibility to obtain the right to use such items and to give proper attribution to the copyright holder. This includes pictures taken by professional photographers and images downloaded from the internet. If you do not hold the copyright for any such item (in whole or part) that is included in your paper, please complete and return this [Third Party Rights Table](http://www.nature.com/documents/thirdpartyrights-origres.doc), and attach any grant of rights that you have collected.

For more information on our licence policy, please consult <http://npg.nature.com/authors>.

ORCID

Nature Microbiology is committed to improving transparency in authorship. As part of our efforts in this direction, we are now requesting that all authors identified as 'corresponding author' create and link their Open Researcher and Contributor Identifier (ORCID) with their account on the Manuscript Tracking System (MTS) prior to acceptance. ORCID helps the scientific community achieve unambiguous attribution of all scholarly contributions. For more information please visit <http://www.springernature.com/orcid>

For all corresponding authors listed on the manuscript, please follow the instructions in the link below to link your ORCID to your account on our MTS before submitting the final version of the manuscript. If you do not yet have an ORCID you will be able to create one in minutes.

IMPORTANT: All authors identified as 'corresponding author' on the manuscript must follow these instructions. Non-corresponding authors do not have to link their ORCIDs but are encouraged to do so. Please note that it will not be possible to add/modify ORCIDs at proof. Thus, if they wish to have their ORCID added to the paper they must also follow the above procedure prior to acceptance.

To support ORCID's aims, we only allow a single ORCID identifier to be attached to one account. If you have any issues attaching an ORCID identifier to your MTS account, please contact the [Platform Support Helpdesk](http://platformsupport.nature.com/).

Nature Research journals [encourage authors to share their step-by-step experimental protocols](https://www.nature.com/nature-research/editorial-policies/reporting-standards#protocols) on a protocol sharing platform of their choice. Nature Research's Protocol Exchange is a free-to-use and open resource for protocols; protocols deposited in Protocol Exchange are citable and can be linked from the published article. More details can found at www.nature.com/protocolexchange/about.

We hope that you will support this initiative and supply the required information. Should you have any query or comments, please do not hesitate to contact me.

We hope to hear from you within two weeks; please let us know if the revision process is likely to take longer.

Final Decision Letter:

Dear Kirsty and Tim,

I am pleased to accept your Article "Characterization of antimicrobial resistant Gram-negative bacteria that cause neonatal sepsis in seven low and middle-income countries" for publication in Nature Microbiology and please accept our apologies for the time it has taken us to contact you with a decision on your manuscript, which is due to our current high submission volume. Thank you for having chosen to submit your work to us and many congratulations.

Before your manuscript is typeset, we will edit the text to ensure it is intelligible to our wide readership and conforms to house style. We look particularly carefully at the titles of all papers to ensure that they are relatively brief and understandable.

Acceptance of your manuscript is conditional on all authors' agreement with our publication policies (see www.nature.com/nmicrobiolate/authors/gta/content-type/index.html). In particular your manuscript must not be published elsewhere and there must be no announcement of the work to any media outlet until the publication date (the day on which it is uploaded onto our website).

Nature Microbiology is a Transformative journal and offers an immediate open access option through payment of an article-processing charge (APC) for papers submitted after 1 January, 2021 . In the event that authors choose to publish under the subscription model, Nature Research allows authors to self-archive the accepted manuscript (the version post-peer review, but prior to copy-editing and typesetting) on their own personal website and/or in an institutional or funder repository where it can be made publicly accessible 6 months after first publication, in accordance with our self-archiving policy. [Please review our self-archiving policy](https://www.nature.com/nature-research/editorial-policies/self-archiving-and-license-to-publish) for more information.

Several funders require deposition the accepted manuscript (AM) to PubMed Central or Europe PubMed Central. To enable compliance with these requirements, Nature Research therefore offers a free manuscript deposition service for original research papers supported by a number of PMC/EPMC participating funders. If you do not choose to publish immediate open access, we can deposit the accepted manuscript in PMC/Europe PMC on your behalf, if you authorise us to do so.

Congratulations once again and I look forward to seeing the article published.